# Multi-Objective Learning for Diffusion Models: A Statistical Theory under Semi-Supervised Learning

**Ziheng Cheng** [* 1]  **Yixiao Huang** [* 1]  **Hanlin Zhu** [1]  **Haoran Geng** [1]  **Somayeh Sojoudi** [1]  **Jitendra Malik** [1]  **Pieter Abbeel** [1]  **Xin Guo** [1]

## Abstract

Diffusion models are increasingly used as powerful conditional generators, yet real deployments often involve multiple target distributions arising from different tasks, e.g., diverse prompt domains in text-to-image generation, or multiple environments in robotics with diffusion policies. This naturally leads to a multi-objective learning (MOL) problem. A key challenge is that achieving good Pareto trade-offs can require a generalist model class with substantially larger capacity than what suffices for solving any individual task, thereby increasing statistical cost since sample complexity typically scales with the model complexity. To reconcile this, we develop a principled MOL framework for diffusion models with limited data: a semi-supervised regime where paired (labeled) samples are scarce, but (unlabeled) condition data are abundant. We propose a two-stage training procedure that first fits lightweight specialist models from limited paired data, and then distills them into a generalist model by generating pseudo-samples. We establish generalization bounds showing that the required number of paired samples only depends on the complexity of the specialist model classes. We further extend the theory to diffusion policies for sequential decision making to account for distribution shift in on-policy rollouts. Extensive experiments on robotic control and image restoration tasks are conducted to verify our theoretical results.

[*]Equal contribution  [1]University of California, Berkeley. Correspondence to: <ziheng_cheng@berkeley.edu>, <yixiaoh@berkeley.edu>.

*Proceedings of the 43rd International Conference on Machine Learning*, Seoul, South Korea. PMLR 306, 2026. Copyright 2026 by the author(s).

## 1. Introduction

Diffusion models have emerged as a versatile paradigm for conditional generation, powering applications such as text-to-image synthesis (Ho et al., 2020; Song et al., 2020; Ho & Salimans, 2022; Rombach et al., 2022), image restoration and inverse problems (Saharia et al., 2022; Kawar et al., 2022), robotic control via diffusion policies (Janner et al., 2022; Chi et al., 2025), and life science (Song et al., 2021; Guo et al., 2024). In many real-world deployments, however, one rarely faces a *single* target distribution. Instead, practitioners often face *multiple* distributions of interest induced by different domains, tasks, or environments. For instance, an image system may be expected to perform robustly across heterogeneous prompt domains (e.g., open-domain prompts vs. technical jargon (Kather et al., 2022)), corruption patterns (e.g., inpainting vs. blurring) and visual styles (e.g., photorealistic vs. anime (Nichol et al., 2021)); while a diffusion policy may need to imitate expert behaviors across multiple environments with different rewards (e.g., goal-reaching vs. safety) and dynamics (e.g., changing camera viewpoints (Ze et al., 2024)). This motivates a *multi-objective learning* (MOL) (Jin, 2007) perspective: learn a single conditional diffusion model that attains Pareto-optimal performance across the target distributions and supports explicit trade-offs among objectives.

However, training a generalist diffusion model typically relies on a large amount of paired data $(x, y)$, where $y$ is the condition and $x$ is the generated variable. In many settings, acquiring paired samples is expensive (e.g., collecting curated paired text–image data, or expert human demonstrations in robotics), whereas obtaining *unpaired conditions* is substantially cheaper (e.g., abundant text prompts, states or observations). This creates an asymmetric regime: we have limited paired samples from each objective, but abundant access to the marginal distribution of conditions, also known as semi-supervised or weak supervision regime.

A central question of this paper is:

*Can we learn a Pareto optimal diffusion model in a sample-efficient manner by leveraging weakly supervised data?*

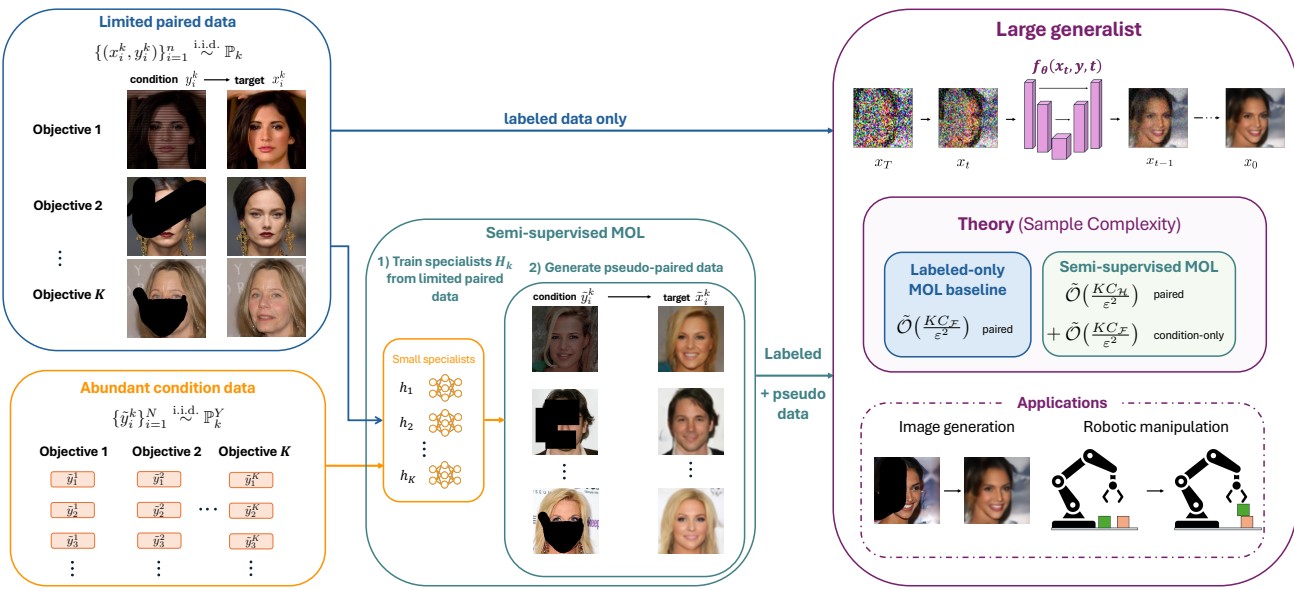

*Figure 1.* Semi-supervised multi-objective learning for conditional diffusion models. Given limited paired data $\{(x_i^k, y_i^k)\}_{i=1}^n \overset{\text{i.i.d.}}{\sim} \mathbb{P}_k$ and abundant condition-only data $\{\tilde{y}_i^k\}_{i=1}^N \overset{\text{i.i.d.}}{\sim} \mathbb{P}_k^Y$ for each objective $k$, we first train lightweight specialists $\widehat{h}_k \in \mathcal{H}_k$ and use them to generate pseudo-paired data. A large diffusion generalist in class $\mathcal{F}$ is then trained on labeled and pseudo-labeled data. This reduces the paired-sample complexity from scaling with the generalist complexity $C_\mathcal{F}$ to scaling with the specialist complexity $C_\mathcal{H}$ where $C_\mathcal{H} \ll C_\mathcal{F}$.

One key observation is that, while training a *generalist* model that performs well across many objectives is statistically expensive, learning a *specialist* model for each individual objective can be far more sample-efficient: the specialist model is typically lightweight with lower capacity, and can be learned from substantially fewer paired samples. For example, in robotics, training a specialist diffusion policy for a single manipulation task (e.g., PUSH-T) can often be done with a relatively small backbone (e.g., a ∼12M-parameter ResNet (Chi et al., 2025)), whereas learning a generalist vision–language–action policy that covers diverse tasks typically demands far more data (e.g., OpenVLA at ∼7B parameters (Kim et al., 2024)).

This motivates a two-stage learning algorithm: we first fit the specialist diffusion models respectively using limited paired data, and then use them to generate pseudo-samples that facilitate learning a generalist model. This procedure is simple, broadly applicable, and aligns with how diffusion models are commonly bootstrapped in practice (You et al., 2023; Li et al., 2024; Xiao et al., 2025). A closely related work (Wegel et al., 2025) has also studied similar semi-supervised learning algorithms for prediction tasks. To our knowledge, however, a comparable theoretical understanding of the statistical rates for multi-objective generative models remains missing.

In this paper, we take the first step towards addressing the central question above and develop a statistical theory for MOL of diffusion models under semi-supervised learning.

Our main contributions are summarized as follows:

- In Section 2, we formalize learning a single diffusion model across multiple target distributions through Pareto optimality, characterized by proper scalarizations. We propose a practical semi-supervised training pipeline that first learns lightweight specialist diffusion models for each objective from limited paired data, and then trains a generalist diffusion model using pseudo-samples generated by the specialists.

- In Section 3.1, we provide nonasymptotic generalization bounds on the score matching objectives and derive distributional estimation guarantees (in total variation distance) under general scalarizations. Notably, our results indicate that required number of paired (labeled) samples depends only on the complexity of the specialist model classes, rather than the large generalist class. In Section 3.2, we further sharpen the rates for linear (weighted-sum) scalarizations via a more refined analysis.

- In Section 3.3, we study diffusion policies in sequential decision making and address the practical distribution shift issue induced by on-policy rollouts. We extend our analysis and establish, to our knowledge, the first theoretical guarantee on the sub-optimality gap of diffusion policies in imitation learning.

Overall, this paper provides a principled MOL framework for diffusion models with limited data, bridging theory and

practice for semi-supervised learning. We also conduct extensive experiments on robotic manipulation and image restoration tasks in Section 5 to verify the theoretical results.

## 2. Preliminaries and Problem Setup

**Notations** We use $x$ and $y$ to denote the data and conditions, respectively. The blackboard bold letter $\mathbb{P}$ represents the joint distribution of $(x, y)$, while the lowercase $p$ denotes its density function. $\mathbb{P}^Y$ stands for the marginal distribution of condition $y$. The notation $\Delta(\Omega)$ denotes the probability simplex over a sample space $\Omega$ and $\Delta^K$ is $(K-1)$-dimensional standard simplex. We use $|x|$ to denote the coordinate-wise absolute value. The norm $\|\cdot\|$ refers to the $\ell_2$-norm for vectors and the spectral norm for matrices, and $\|\cdot\|_p$ denotes general $\ell_p$-norm. Finally, we use standard $\lesssim, \mathcal{O}(\cdot)$ to omit constant factors and $\widetilde{\mathcal{O}}(\cdot)$ to omit logarithmic factors.

### 2.1. Conditional Diffusion Models

Let $\mathbb{R}^{d_x}$ denote the data space and $[0, 1]^{d_y}$ denote the condition space. Let $\mathbb{P}$ be any joint distribution over $\mathbb{R}^{d_x} \times [0, 1]^{d_y}$ with density $p$ and $\mathbb{P}(\cdot|y)$ be the conditional distribution with density $p(\cdot|y)$. As in diffusion models, the forward process is defined as an Ornstein–Uhlenbeck (OU) process,

$$\mathrm{d}X_t = -X_t\mathrm{d}t + \sqrt{2}\mathrm{d}W_t, X_0 \sim \mathbb{P}(\cdot|y),$$

where $\{W_t\}_{t\geq 0}$ is a standard Wiener process. We denote the distribution of $X_t$ as $\mathbb{P}_t(\cdot|y)$. Note that the limiting distribution $\mathbb{P}_\infty(\cdot|y)$ is a standard Gaussian $\mathcal{N}(0, I)$.

To generate new samples, we can reverse the forward process from any $T > 0$ and $X_0^\leftarrow \sim \mathbb{P}_T(\cdot|y)$:

$$\mathrm{d}X_t^\leftarrow = (X_t^\leftarrow + 2\nabla \log p_{T-t}(X_t^\leftarrow|y))\mathrm{d}t + \sqrt{2}\mathrm{d}\overline{W}_t. \quad (2.1)$$

Alternatively one may replace Eq. (2.1) by probability flow ODE (Song et al., 2020)

$$\mathrm{d}X_t^\leftarrow = (X_t^\leftarrow + \nabla \log p_{T-t}(X_t^\leftarrow|y))\mathrm{d}t. \quad (2.2)$$

In general, one does not have access to the exact conditional score function $\nabla \log p_{T-t}$, which needs to be estimated. For any $(x, y) \sim \mathbb{P}$ and a score estimator $s$, define the individual denoising score matching objective (Vincent, 2011) with early stopping as

$$\ell(x, y, s) := \mathbb{E}_{t, x_t|x}\big[\|s(x_t, y, t) - \nabla \log \phi_t(x_t|x)\|^2\big],$$

where $t \sim \mathrm{Unif}([T_0, T])$, and $\phi_t(x_t|x) = \mathcal{N}(x_t|\alpha_t x, \sigma_t^2 I)$ with $\alpha_t = e^{-t}$ and $\sigma_t^2 = 1 - e^{-2t}$ is the transition kernel of $x_t|x_0 = x$. Note that here $T_0$ is an early-stopping time. The population error of score matching is

$$L_\mathbb{P}(s) := \mathbb{E}_{(x,y)\sim\mathbb{P}}\mathbb{E}_{t,x_t}[\|s(x_t, y, t) - \nabla \log p_t(x_t|y)\|^2]$$
$$= \mathbb{E}_{(x,y)\sim\mathbb{P}}[\ell(x, y, s) - \ell(x, y, s_\mathbb{P}^*)].$$

Here $s_\mathbb{P}^*$ denotes the true score function.

With a score estimator $\widehat{s}$, the generative process is to simulate from $\widehat{X}_0^\leftarrow \sim \mathcal{N}(0, I)$ and replace the score function in Eq. (2.1) or Eq. (2.2) with $\widehat{s}$. Given the more desirable theoretical properties of SDE sampler over its ODE alternatives, our analysis focuses exclusively on Eq. (2.1) and we denote the resulting conditional distribution by $\mathbb{P}_{\widehat{s}}(\cdot|y)$.

### 2.2. Multi-Objective Learning

Let $\mathbb{P}_1, \cdots, \mathbb{P}_K$ be $K$ target distributions of interest over $\mathbb{R}^{d_x} \times [0, 1]^{d_y}$. We denote the ground truth score function for each distribution by $s_k^*(x, y, t)$. Given a score function class $\mathcal{F}$, we aim to learn a single estimator $f \in \mathcal{F}$ and the corresponding diffusion model $\mathbb{P}_f$ that performs well simultaneously across all $K$ conditional distributions. Concretely, we seek for the *Pareto optimal* set defined as follows.

**Definition 2.1** (Pareto optimality). Given discrepancies $\{d_k(\cdot, \cdot)\}_{k=1}^K$ between two distributions, we say a score function $f \in \mathcal{F}$ is Pareto optimal in $\mathcal{F}$ if there does not exist $f' \in \mathcal{F}$ such that, for all $k \in [K]$,

$$\mathbb{E}_{y\sim\mathbb{P}_k^Y}[d_k(\mathbb{P}_{f'}(\cdot|y), \mathbb{P}_k(\cdot|y))] \leq \mathbb{E}_{y\sim\mathbb{P}_k^Y}[d_k(\mathbb{P}_f(\cdot|y), \mathbb{P}_k(\cdot|y))],$$

with strict inequality for at least one $k$.

Here $d_k(\cdot, \cdot)$ can be chosen as a distributional distance when the goal is distribution estimation, or a performance metric such as sub-optimality gap in the context of sequential decision making.

In fact, each Pareto optimal model corresponds to a distinct trade-off among the $K$ objectives and such trade-offs can be characterized by scalarization functions $\mathcal{S} : \mathbb{R}^K \to \mathbb{R}$,

$$f_\mathcal{S}^* = \arg\min_{f\in\mathcal{F}} \mathcal{S}\big(\big(\mathbb{E}_{y\sim\mathbb{P}_k^Y}[d_k(\mathbb{P}_f(\cdot|y), \mathbb{P}_k(\cdot|y))]\big)_k\big). \quad (2.3)$$

Following Wegel et al. (2025), we assume $\mathcal{S}$ satisfies (i) reverse triangle inequality and (ii) positive homogeneity:

$$|\mathcal{S}(\vec{u}) - \mathcal{S}(\vec{v})| \leq \mathcal{S}(|\vec{u} - \vec{v}|), \mathcal{S}(\alpha\vec{u}) = \alpha\mathcal{S}(\vec{u}), \forall \vec{u}, \vec{v}, \alpha \geq 0.$$

Common scalarization methods include linear combination $\mathcal{S}(\vec{u}) = \sum_k \lambda_k \vec{u}_k$ for $\lambda \in \Delta^K$, Chebyshev aggregation $\mathcal{S}(\vec{u}) = \max_k \vec{u}_k$ and general $\ell_p$-norm $\mathcal{S}(\vec{u}) = \|u\|_p$. More examples can be found in Ehrgott (2005); Miettinen (1999).

### 2.3. Two-Stage Semi-supervised Training

As discussed earlier, in order to achieve reasonable Pareto trade-offs across multiple objectives, the *generalist* function class $\mathcal{F}$ must be sufficiently expressive–often larger than what is necessary to solve any single task in isolation. If we assume that for each $k$, there is a lightweight *specialist* score function class $\mathcal{H}_k \subset \mathcal{F}$, i.e., a restricted hypothesis space tailored to objective $\mathbb{P}_k$, then learning a CDM $h_k \in \mathcal{H}_k$

that approximates $\mathbb{P}_k$ would be relatively easy and sample-efficient, as $\mathcal{H}_k$ has substantially smaller complexity than the full class $\mathcal{F}$. This motivates a two-stage semi-supervised learning strategy: we first fit the specialist models $\{\mathbb{P}_{h_k}\}_{k=1}^K$ respectively, which are then distilled into a generalist model by generating pseudo-samples, as illustrated in Figure 1.

Mathematically, for each $k \in [K]$, suppose we have limited pairs $\{(x_i^k, y_i^k)\}_{i=1}^n \overset{\text{i.i.d.}}{\sim} \mathbb{P}_k$ and abundant condition samples $\{\widetilde{y}_i^k\}_{i=1}^N \overset{\text{i.i.d.}}{\sim} \mathbb{P}_k^Y$ with $N \gg n$. In the first stage, we train a specialist model for each task using paired data:

$$\widehat{h}_k := \underset{h_k \in \mathcal{H}_k}{\arg\min} \frac{1}{n} \sum_{i=1}^n \ell(x_i^k, y_i^k, h_k). \qquad (2.4)$$

In the second stage, given a scalarization $\mathcal{S}$, we generate pseudo-samples $\widetilde{x}_i^k$ from $\mathbb{P}_{\widehat{h}_k}(\cdot|\widetilde{y}_i^k)$ and then train the generalist model:

$$\widehat{f}_{\mathcal{S}} := \underset{f \in \mathcal{F}}{\arg\min} \, \mathcal{S}(\widetilde{L}_1(f), \cdots, \widetilde{L}_K(f)),$$

$$\widetilde{L}_k(f) := \frac{1}{N} \sum_{i=1}^N [\ell(\widetilde{x}_i^k, \widetilde{y}_i^k, f) - \ell(\widetilde{x}_i^k, \widetilde{y}_i^k, \widehat{h}_k)]. \qquad (2.5)$$

**Examples** Below we present two typical real-world applications of semi-supervised MOL setup in CDMs.

- Image generation: A conditional diffusion model learns $p(x|c)$, where $c$ denotes a text prompt or an image to be edited. Multiple target distributions arise naturally from heterogeneous data sources, e.g., different text domains, image styles or human preferences. The goal is to train a single conditional model capable of fitting these heterogeneous data distributions simultaneously. In practice, text prompts or corrupted images are easy to collect at scale from web, whereas high-quality paired text/image–image annotations are scarce and costly to curate (Wang et al., 2023).

- Diffusion policies: For robotic control, a diffusion policy $\pi(a|s)$ models the action distribution conditioned on state (Chi et al., 2025). In multi-task or goal-conditioned robotics, each environment (with different rewards or dynamics) and corresponding optimal policy induce a distinct state–action distribution $\mathbb{P}(s, a)$. The multi-objective learner must capture the diverse optimal policies within a unified policy model. Expert demonstrations providing paired $(s, a)$ supervision are expensive to collect (Xiao et al., 2025).

## 3. Main Results

In this section, we present a statistical theory of semi-supervised learning algorithm for MOL in diffusion models.

All formal statements and detailed proofs are provided in Appendix B.

Throughout this paper, we make the following standard and mild regularity assumptions on the data distribution $\mathbb{P}$ and score function classes.

**Assumption 3.1** (Sub-gaussian tail)**.** *For any $k \in [K]$, $\mathbb{P}_k$ is supported on $\mathbb{R}^{d_x} \times [0, 1]^{d_y}$ and admits a continuous density $p_k(x, y) \in \mathcal{C}^2(\mathbb{R}^{d_x} \times [0, 1]^{d_y})$. Moreover, the conditional distribution $p_k(x|y) \le C_1 \exp(-C_2\|x\|^2)$ for some constant $C_1, C_2$.*

**Assumption 3.2** (Lipschitz score)**.** *For any $h \in \cup_k \mathcal{H}_k \cup \mathcal{F}$, $h$ has linear growth in $x$, i.e., $\|h(x, y, t)\| \le M_0 + M_1\|x\|$ for some $M_0, M_1 \ge 1$.*

In fact, our analysis can be seamlessly extended to more general polynomial growth conditions of the form $\|h(x, y, t)\| \le \mathbf{poly}(\|x\|)$. We adopt the linear-growth assumption here for ease of exposition. Moreover, as shown in Cheng et al. (2025, Lemma 3.1), this assumption holds for ground truth score $s_k^*(x, y, t)$ as long as $\nabla \log p_k(x|y)$ is Lipschitz, which is a standard and mild regularity assumption in the literature (Chen et al., 2022b; 2023a). Following Wegel et al. (2025), we also impose the following realizability assumption for the specialist function classes.

**Assumption 3.3** (Realizability)**.** *The ground truth score function $s_k^* \in \mathcal{H}_k$ for any $k \in [K]$.*

This assumption is crucial because pseudo samples are drawn from the learned specialists in the second stage; if a specialist class is misspecified, the induced distribution $\mathbb{P}_{\widehat{h}_k}$ may be systematically biased away from $\mathbb{P}_k$, and this bias would propagate to the generalist training, introducing an irreducible error that is not controlled by sample size.

To quantify the complexity of a score function class, we introduce the following notion of covering number:

**Definition 3.1** (Covering number)**.** Let $(\Psi, \|\cdot\|)$ be a metric space of functions. We say $\Psi_0 \subseteq \Psi$ is a $\rho$-net of $\Psi$ if for any $\psi \in \Psi$, there exists $\psi_0 \in \Psi_0$ such that $\|\psi - \psi_0\| \le \rho$. The covering number $\mathcal{N}(\Psi, \|\cdot\|, \rho)$ is defined as the minimal cardinality of a $\rho$-net of $\Psi$.

For the score function classes $\mathcal{F}, \{\mathcal{H}_k\}_k$, we consider the $L^\infty$-metric on a compact domain $\Omega_R := [-R, R]^{d_x} \times [0, 1]^{d_y} \times [T_0, T]$ and write $\|\cdot\|_{L^\infty(\Omega_R)}$ accordingly. Here the truncation radius is taken as $R = \mathcal{O}(\log(NK))$ (see formal derivation in the proof). In the sequel we abbreviate $\log \mathcal{N}(\mathcal{F}, \rho) = \log \mathcal{N}(\mathcal{F}, \|\cdot\|_{L^\infty(\Omega_R)}, \rho)$ and use the analogous shorthand for $\{\mathcal{H}_k\}_k$ for notational simplicity. We remark that for commonly used neural network families $\Psi$ with $L^\infty$-metric, the log-covering number typically scales as $\log \mathcal{N}(\Psi, \rho) \le \mathcal{C}_\Psi \log(1/\rho)$ and $\mathcal{C}_\Psi$ is a complexity parameter that scales with the effective size of the architecture (and in particular is closely tied to the number of trainable

parameters and depth/width). For instance, for an $L$-layer MLP of width $W$, one can upper bound the complexity by $\mathcal{C}_{\mathrm{MLP}} \lesssim LW^2 \log(LW)$ (Chen et al., 2022a), while for a Transformer with $L$ layers, $M$ attention heads, and embedding dimension $D$, a typical bound yields $\mathcal{C}_{\mathrm{TF}} \lesssim ML^2D^2$ (Lin et al., 2023).

### 3.1. MOL for Conditional Diffusion Models

Since the pseudo-samples are generated by the specialist model $\mathbb{P}_{\widehat{h}_k}$, we first need to show that $\mathbb{P}_{\widehat{h}_k}$ can approximate each individual target distribution $\mathbb{P}_k$ using paired samples in the first stage.

**Lemma 3.1** (Informal version of Lemma B.5). *It holds with probability no less than $1 - \delta$ that for any $k \in [K]$,*

$$L_{\mathbb{P}_k}(\widehat{h}_k) \lesssim \frac{\log \mathcal{N}(\mathcal{H}_k, \frac{1}{n^2}) + \log(K/\delta)}{n}. \quad (3.1)$$

Let $\mathcal{E}_{\mathcal{S}}(f) := \mathcal{S}(L_{\mathbb{P}_1}(f), \cdots, L_{\mathbb{P}_K}(f))$. Now we are ready to present our first main result for the sample efficiency of semi-supervised learning. In particular, we show that the number of paired samples depends only on the complexity of the specialist model classes, instead of the large generalist class.

**Theorem 3.2** (Informal version of Thm. B.6). *The following holds with probability no less than $1 - \delta$: for any scalarization $\mathcal{S}$, $\mathcal{E}_{\mathcal{S}}(\widehat{f}_{\mathcal{S}}) - \inf_{f \in \mathcal{F}} \mathcal{E}_{\mathcal{S}}(f) \leq \mathcal{S}(\varepsilon_1, \cdots, \varepsilon_K)$, where*

$$\varepsilon_k \lesssim \sqrt{\frac{\log \mathcal{N}(\mathcal{H}_k, \frac{1}{n^2}) + \log(\frac{K}{\delta})}{n}} + \sqrt{\frac{\log \mathcal{N}(\mathcal{F}, \frac{1}{N})}{N}}. \quad (3.2)$$

To interpret this result, following the earlier discussion and assuming the covering numbers satisfy $\log \mathcal{N}(\mathcal{F}, \rho) \leq \mathcal{C}_{\mathcal{F}} \log(1/\rho)$, $\log \mathcal{N}(\mathcal{H}_k, \rho) \leq \mathcal{C}_{\mathcal{H}} \log(1/\rho)$. Then the above generalization bound shows that in order to achieve $\varepsilon_k \leq \varepsilon$, our semi-supervised procedure requires in total $\widetilde{\mathcal{O}}\left(\frac{K\mathcal{C}_{\mathcal{H}}}{\varepsilon^2}\right)$ paired (labeled) samples $(x, y)$ and $\widetilde{\mathcal{O}}\left(\frac{K\mathcal{C}_{\mathcal{F}}}{\varepsilon^2}\right)$ (unlabeled) conditioning variables $y$. It suggests that paired-sample complexity is determined solely by the specialist model classes, not by the capacity of the generalist class. This is similar to the findings for prediction tasks in Wegel et al. (2025). In contrast, vanilla (fully supervised) training requires $\widetilde{\mathcal{O}}\left(\frac{K\mathcal{C}_{\mathcal{F}}}{\varepsilon^2}\right)$ labeled samples to achieve the same accuracy (Zhang et al., 2024, Thm. 2). Therefore, the semi-supervised learning method substantially improves the sample-efficiency whenever $\mathcal{C}_{\mathcal{F}} \gg \mathcal{C}_{\mathcal{H}}$.

**Remark 1.** *Wegel et al. (2025) also studies sample efficiency in semi-supervised learning for prediction tasks. However, their analysis focuses on Bregman losses on bounded domain whereas our setting involves the more intricate score matching objectives in diffusion models. Technically, in Wegel et al. (2025, Thm. 1), their results require*

$\widetilde{\mathcal{O}}(\frac{K\mathcal{C}_{\mathcal{H}}}{\varepsilon^4})$ *labeled samples,* $\widetilde{\mathcal{O}}(\frac{K\mathcal{C}_{\mathcal{F}}}{\varepsilon^2})$ *unlabeled samples to achieve $\varepsilon$-error on Bregman loss. Although learning diffusion models is a more challenging generative problem than prediction, our results yield a sharper labeled-sample complexity by exploiting the specific structure of the score matching objective.*

To further obtain an error bound on distribution estimation, we introduce the following lemma.

**Lemma 3.3** (Informal version of Lemma B.3). *With proper configurations of $T, T_0$, it holds that*

$$\mathbb{E}_{y \sim \mathbb{P}_k^Y} \mathrm{TV}(\mathbb{P}_{\widehat{h}_k}(\cdot|y), \mathbb{P}_k(\cdot|y)) \lesssim \sqrt{L_{\mathbb{P}_k}(\widehat{h}_k)}. \quad (3.3)$$

This lemma provides the key bridge from the score matching objective to distribution estimation.

Next, define the approximation error of the generalist model class as

$$\varepsilon_{\mathrm{apx}}^2(\mathcal{S}) = \inf_{f \in \mathcal{F}} \mathcal{S}\left((\mathbb{E}_{(x,y) \sim \mathbb{P}_k} \mathbb{E}_{t,x_t} \|(f - s_k^*)(x_t, y, t)\|^2)_k\right). \quad (3.4)$$

**Corollary 3.4.** *If further assuming $\mathcal{S}(|\cdot|)$ is coordinatewise non-decreasing and $|\mathcal{S}(\vec{u})| \leq \|\vec{u}\|_\infty$, then*

$$\mathcal{S}\left((\mathbb{E}_{y \sim \mathbb{P}_k^Y} \mathrm{TV}(\mathbb{P}_{\widehat{f}_{\mathcal{S}}}(\cdot|y), \mathbb{P}_k(\cdot|y)))_k\right) \lesssim \varepsilon_{apx}(\mathcal{S}) + \sqrt{\mathcal{S}((\varepsilon_k)_k)},$$

*where $\varepsilon_k$ is bounded by Eq. (3.2).*

Corollary 3.4 converts our statistical guarantee for score matching into distribution estimation in total variation distance. This metric is central to our goal, as it ensures that the learned diffusion model $\mathbb{P}_{\widehat{f}_{\mathcal{S}}}$ can achieve Pareto optimal performance when approximating the $K$ target conditional distributions. We remark that these additional regularity assumptions on $\mathcal{S}$ are mild and satisfied by the commonly used scalarizations mentioned in Section 2.2.

### 3.2. Faster Rate for Linear Scalarizations

The preceding analysis establishes a uniform bound for all possible scalarization methods, yielding a $\widetilde{\mathcal{O}}(n^{-1/4} + N^{-1/4})$ statistical rate in total variation distance. In this section, we show that the bound can be further improved when restricting to *linear* scalarizations. Formally, let $\mathcal{S}_{\mathrm{lin}} := \{\mathcal{S}(\vec{u}) = \sum_{k=1}^K \lambda_k \vec{u}_k : \lambda \in \Delta^K\}$.

**Theorem 3.5** (Informal version of Thm. B.8). *Assume that $\mathcal{F}$ is convex. Then the following holds with probability no less than $1 - \delta$: for any $\mathcal{S} \in \mathcal{S}_{\mathrm{lin}}$ with $\mathcal{S}(\vec{u}) = \sum_{k=1}^K \lambda_k \vec{u}_k$,*

$$\mathcal{S}\left((\mathbb{E}_{y \sim \mathbb{P}_k^Y} \mathrm{TV}(\mathbb{P}_{\widehat{f}_{\mathcal{S}}}(\cdot|y), \mathbb{P}_k(\cdot|y)))_k\right) \lesssim \bar{\varepsilon}_{apx}(\mathcal{S})$$

$$+ \sqrt{\frac{\log \mathcal{N}(\mathcal{F}, \frac{1}{N})}{N}} + \sum_{k=1}^K \lambda_k \sqrt{\frac{\log \mathcal{N}(\mathcal{H}_k, \frac{1}{n^2}) + \log(\frac{K}{\delta})}{n}}.$$

*where the approximation error is defined as*

$$\bar{\varepsilon}_{apx}^2(\mathcal{S}) :=$$
$$\sup_{h_k \in \mathcal{H}_k} \inf_{f \in \mathcal{F}} \mathcal{S}((\mathbb{E}_{y \sim \mathbb{P}_k^Y, x \sim \mathbb{P}_{h_k}(\cdot|y)} \mathbb{E}_{x_t} \|(f - h_k)(x_t, y, t)\|^2)_k).$$

The generalization bound above implies that to achieve $\varepsilon$-excess risk in total variation distance, it suffices to use $\widetilde{\mathcal{O}}(\frac{KC_{\mathcal{H}}}{\varepsilon^2})$ paired samples and $\widetilde{\mathcal{O}}(\frac{KC_{\mathcal{F}}}{\varepsilon^2})$ conditions, improving upon Corollary 3.4. In particular, Thm. 3.5 provides a faster rate for the linear scalarization family and convex function class $\mathcal{F}$.

Note that the approximation error here is a universal bound for all $h_k \in \mathcal{H}_k$. This is strictly stronger than Eq. (3.4) which only requires the approximation capacity of $\mathcal{F}$ for ground truth $s_k^*$. The reason is mainly technical. Recall that the unified model $\widehat{f}_{\mathcal{S}}$ is essentially trained to optimize $\mathcal{S}\big(\big(\mathbb{E}_{y \sim \mathbb{P}_k^Y, x \sim \mathbb{P}_{\widehat{h}_k}(\cdot|y)} \mathbb{E}_{x_t} \|(f - \widehat{h}_k)(x_t, y, t)\|^2\big)_k\big)$, which is lower bounded by the approximation error of $\mathcal{F}$ for arbitrary $\widehat{h}_k \in \mathcal{H}_k$. Although Lemma 3.1 guarantees that $L_{\mathbb{P}_k}(\widehat{h}_k) = \mathbb{E}_{y \sim \mathbb{P}_k^Y, x \sim \mathbb{P}_k(\cdot|y)} \|\widehat{h}_k(x_t, y, t) - s_k^*(x_t, y, t)\|^2$ is small and hence $\mathbb{E}_{y \sim \mathbb{P}_K^Y} \text{TV}(\mathbb{P}_{\widehat{h}_k}(\cdot|y), \mathbb{P}_k(\cdot|y))$ is small, it remains nontrivial to transfer this guarantee to the *on-model* expectation $\mathbb{E}_{y \sim \mathbb{P}_k^Y, x \sim \mathbb{P}_{\widehat{h}_k}(\cdot|y)} \|\widehat{h}_k(x_t, y, t) - s_k^*(x_t, y, t)\|^2$. This is because the $L^2$-accurate score estimator can only imply small $\text{TV}(\mathbb{P}_{\widehat{h}_k}(\cdot|y), \mathbb{P}_k(\cdot|y))$ (Wibisono & Yingxi Yang, 2022); without further regularity assumptions on data distributions, it cannot control stronger density-ratio divergences such as chi-square divergence, which is necessary to replace $\mathbb{P}_k(\cdot|y)$ by $\mathbb{P}_{\widehat{h}_k}(\cdot|y)$ inside the expectation. We hope future works can address this issue.

We notice that Wegel et al. (2025, Thm. 2) also derives improved sample complexity bound via localization techniques, but under stronger structural assumptions on the function classes—in particular, each $\mathcal{H}_k$ is required to be star-shaped around the ground-truth score $s_k^*$. In contrast, our analysis in Thm. 3.5 does not impose such a restriction and yields more general guarantees for distribution estimation, rather than prediction error.

### 3.3. MOL for Diffusion Policies: Tackling Distribution Shift in Sequential Decision Making

In the previous section, we assume access to conditions $\{\widehat{y}_i^k\} \overset{\text{i.i.d.}}{\sim} \mathbb{P}_k^Y$. In broader settings such as diffusion policies in imitation learning, this assumption may fail: the distribution of conditioning variables is induced by the learned policy itself, resulting in a distribution shift in the condition variables and necessitating a more delicate analysis. In this section, we show that our semi-supervised learning framework can accommodate this issue and still yield improved sample-complexity guarantees.

We present the setup of diffusion policies as follows. Let $\mathcal{M}$ be the space of decision-making environments, where each $M \in \mathcal{M}$ is an infinite horizon Markov Decision Process (MDP) sharing the same state space $S$, action space $A$, discount factor $\gamma$. And each $M \in \mathcal{M}$ has its own transition kernel $\mathcal{T}_M : S \times A \to \Delta(S)$, reward function $r_M : S \times A \to [-1, 1]$, and initial distribution $\rho_M \in \Delta(S)$. The policy is defined as a map $\pi : S \to \Delta(A)$, and the value function of $M$ under policy $\pi$ is

$$V_M(\pi) := \mathbb{E}_{a_t \sim \pi(\cdot|s_t), s_{t+1} \sim \mathcal{T}_M(\cdot|s_t, a_t)} \Big[ \sum_{t=0}^{\infty} \gamma^t r_M(s_t, a_t) \Big].$$

Denote the visitation measure of state-action as $\mathbb{P}_M^\pi(s, a) := (1 - \gamma)\mathbb{E}[\sum_{t=0}^{\infty} \gamma^t \mathbb{P}(s_t = s|\pi)\pi(a|s)]$. Suppose that there are $K$ environments $(M_k)_{k \in [K]} \in \mathcal{M}$ with expert policy $(\pi_k^*)_{k \in [K]}$. The diffusion policy is a conditional diffusion model that takes a state $s$ as the conditioning variable and generates a (stochastic) action $a$. To unify the notation, let $x = a, y = s, \mathbb{P}_k := \mathbb{P}_{M_k}^{\pi_k^*}$ and assume $A = \mathbb{R}^{d_x}, S = [0, 1]^{d_y}$.

In the first stage, for each environment $M_k$, we are given a limited set of expert demonstrations, i.e. $\{(s_i^k, a_i^k)\}_{i=1}^n \overset{\text{i.i.d.}}{\sim} \mathbb{P}_{M_k}^{\pi_k^*}$ and train a diffusion policy $\pi_{\widehat{h}_k}$. However, in the MDP setting, it is impractical to collect a large number of states $\{\widetilde{s}_i^k\}_{i=1}^N$ sampled from the expert trajectory $\mathbb{P}_{M_k}^{\pi_k^*}$ as it requires rolling out expert policy and is of comparable cost to collecting state-action pairs. Therefore, in the second stage, we directly simulate trajectories using the learned $\pi_{\widehat{h}_k}$ under environment $M_k$ to collect $\{(\widetilde{s}_i^k, \widetilde{a}_i^k)\}_{i=1}^N \overset{\text{i.i.d.}}{\sim} \mathbb{P}_{M_k}^{\pi_{\widehat{h}_k}}$, after which we apply the same training procedures as before to obtain the generalist policy $\pi_{\widehat{f}_{\mathcal{S}}}$.

This pipeline introduces a *distribution shift* between expert-generated data and on-policy rollouts, a phenomenon that is inherent to the MDP setting and well known in imitation learning (Ross & Bagnell, 2010; Foster et al., 2024). Regardless of the difference, we follow the settings in Thm. 3.2 and bound the sub-optimality gap as follows.

**Theorem 3.6** (Informal version of Thm. B.9). *Under the MDP setting, assume that $\mathcal{F}$ is convex. The following holds with probability at least $1 - \delta$: for any $\mathcal{S} \in \mathcal{S}_{lin}$ with $\mathcal{S}(\vec{u}) = \sum_{k=1}^K \lambda_k \vec{u}_k$,*

$$\mathcal{S}\big((V_{M_k}(\pi_k^*) - V_{M_k}(\pi_{\widehat{f}_{\mathcal{S}}}))_k\big) \lesssim (1 - \gamma)^{-2}\bar{\varepsilon}_{apx}(\mathcal{S})$$
$$+ \sqrt{\frac{\log \mathcal{N}(\mathcal{F}, \frac{1}{N})}{N}} + \sum_{k=1}^K \lambda_k \sqrt{\frac{\log \mathcal{N}(\mathcal{H}_k, \frac{1}{n^2}) + \log(\frac{K}{\delta})}{n}}.$$

This highlights that semi-supervised learning algorithm can successfully address the distribution shift issue while retaining sample efficiency. Similar to our earlier results, the

labeled-sample complexity only depends on the complexity of specialist model classes $\{\mathcal{H}_k\}_k$ rather than the generalist model class $\mathcal{F}$. To our knowledge, Thm. 3.6 is the first theoretical guarantee on the sub-optimality gap of diffusion policies in imitation learning.

**Remark 2.** *In the current state-of-the-art analysis for single-objective imitation learning with a policy class of finite cardinality (Foster et al., 2024), the sub-optimality gap typically scale as $\widetilde{\mathcal{O}}(\frac{\mathcal{C}_\mathcal{F}}{N})$. In contrast, when reduced to the single-objective setting, our diffusion-policy result yields a rate of $\widetilde{\mathcal{O}}(\sqrt{\frac{\mathcal{C}_\mathcal{F}}{N}})$. The gap stems from the conversion step in Lemma 3.3, which controls total variation distance by the square root of the score-matching error. As a consequence, even if the score-matching objective enjoys a fast rate with $\widetilde{\mathcal{O}}(\frac{\mathcal{C}_\mathcal{F}}{N})$, the induced total variation bound inherits an additional square root and thus becomes slower. We believe that a more fine-grained analysis of the score-to-optimality translation for diffusion policies may close this gap in the future.*

## 4. Related Works

Recently, the score approximation theory of deep neural network families and corresponding statistical rates for diffusion models have been developed (Oko et al., 2023; Chen et al., 2023a; Wibisono et al., 2024; Hu et al., 2024). Further extension of this framework to conditional diffusion models with classifier-free guidance is studied in Fu et al. (2024), and Cheng et al. (2025) analyzes the sample efficiency of transfer learning for diffusion models and diffusion policies across multiple source tasks. For a given a bound on the score matching error, there are additional works establishing the convergence rate of discrete samplers for diffusion models (Chen et al., 2022b; Lee et al., 2023; Chen et al., 2023b). By combining score estimation and sampler discretization analyses, these results yield end-to-end bounds on distribution estimation error.

The statistical theory of multi-objective learning (MOL) has been studied primarily in the fully supervised setting. For example, Haghtalab et al. (2022) proposes an adaptive sampling strategy for multi-distribution learning under Chebyshev (min–max) scalarization, and Zhang et al. (2024) further develops an improper learning algorithm achieving optimal sample complexity. In contrast, semi-supervised MOL has remained far less understood. In particular, it is unclear if and how unlabeled data can improve labeled-sample efficiency. The closest work to ours is Wegel et al. (2025), which shows that semi-supervised learning can reduce labeled-sample complexity for *prediction* problems under Bregman losses; we refer readers to Wegel et al. (2025) for a broader discussion of related MOL theory. To the best of our knowledge, analogous guarantees for *generative* diffusion models, specifically for distribution estimation under

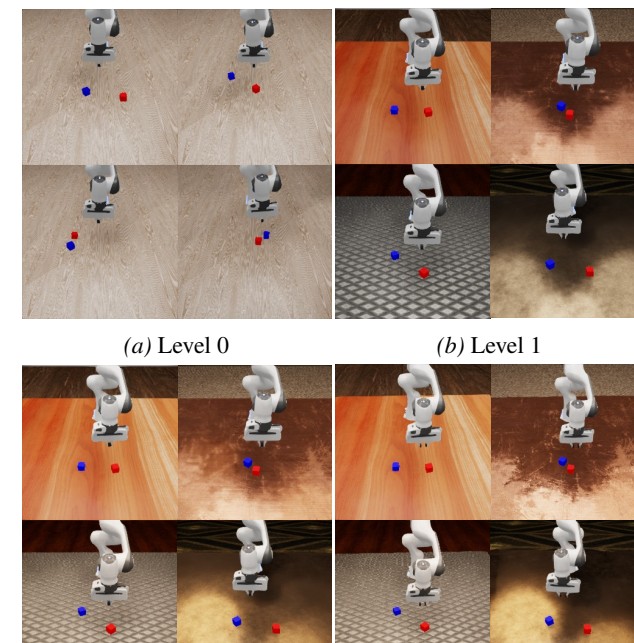

*(a)* Level 0      *(b)* Level 1

*(c)* Level 2      *(d)* Level 3

*Figure 2.* Visualization of the domain-randomization levels used in the robotics manipulation experiments. Level 0 uses the canonical scene, Level 1 adds scene/material randomization, Level 2 further adds lighting randomization, and Level 3 additionally introduces camera-pose randomization. Levels 2–3 are used as held-out OOD evaluations.

weak supervision, have not been established.

The paradigm of distilling specialized experts into a single generalist policy has been studied extensively in the literature (Levine & Abbeel, 2014), particularly in the context of robot foundation models. Recent approaches such as RLDG (Xu et al., 2024) and PLD (Xiao et al., 2025) demonstrate that rollouts generated by lightweight, RL-trained specialists or residual actors can be used to fine-tune generalist policies, often outperforming those trained based on human demonstrations. Compared with prior works that establish the empirical effectiveness of policy distillation, we formalize the "specialist-to-generalist" pipeline within a MOL framework, where the goal is to achieve Pareto optimality across diverse tasks. Crucially, our results provide the first niformal justification for why distilling specialists is often more sample-efficient than direct multi-task learning.

## 5. Experiments

Our experiments empirically validate the two-stage semi-supervised approach to multi-objective learning (MOL) introduced in the previous sections. Specifically, we ask:

1. Does augmenting the labeled split with specialist-

*Table 1.* Robotic manipulation results on StackCube and PickCube. We report closed-loop success rates over 100 test trajectories. Levels 0–1 are in-distribution evaluations, while Levels 2–3 are out-of-distribution (OOD) evaluations with additional domain randomization.

| Task | Level | Semi-supervised MOL (ours) | Labeled-only MOL baseline |
|---|---|---|---|
| *StackCube* | Level 0 | **43%** | 36% |
| | Level 1 | **44%** | 36% |
| | Level 2 (OOD) | **40%** | 24% |
| | Level 3 (OOD) | **22%** | 8% |
| *PickCube* | Level 0 | **69%** | 25% |
| | Level 1 | **49%** | 26% |
| | Level 2 (OOD) | **41%** | 35% |
| | Level 3 (OOD) | **42%** | 26% |

generated pseudo-samples outperform the labeled-only multi-task MOL baseline?

2. Does the same recipe apply across qualitatively different domains and tasks?

We instantiate our approach in two domains: robotic manipulation in simulation (Section 5.1) and image restoration on natural images (Section 5.2). In both settings, per-task specialists are trained only on the labeled split and then used to generate pseudo-samples on unlabeled conditions. A larger generalist is trained on the union of labeled and pseudo-labeled data, while the labeled-only MOL baseline uses the same generalist architecture but only the labeled split. Full training details are deferred to Section A.

## 5.1. Robotic Manipulation

We evaluate on StackCube and PickCube (Gu et al., 2023) in RoboVerse (Geng et al., 2025) on top of Isaac Sim (NVIDIA, 2024a). Each family is instantiated under three *domain-randomization (DR)* elements that we treat as the multiple objectives:

- **Scene/material randomization** of the table, ground, and wall, drawn from curated subsets of ARNOLD (Gong et al., 2023) and vMaterials (NVIDIA, 2024b);

- **Lighting randomization** of intensity, color temperature, and source geometry;

- **Camera-pose randomization**, where a fraction of cameras are repositioned to side-facing angles.

We compose them into four cumulative levels: L0 (canonical scene), L1 (adds scene/material), L2 (adds lighting on top of L1), and L3 (adds camera-pose). L1 is itself instantiated as three single-element variants—L1-Table, L1-Floor, L1-Wall—which together with L0 yield four training variants per family; L2 and L3 are held out as OOD (see Figure 2).

All policies are CNN-based diffusion policies (Chi et al., 2025) with a DDPM (Ho et al., 2020) U-Net1D action

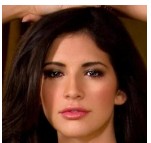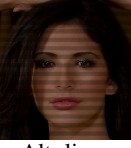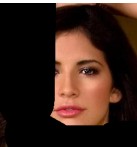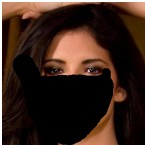

| Clean target | Alt. lines | Wide mask | Face mask |

*Figure 3.* Visualization of the CelebA-HQ inpainting objectives. Each objective corresponds to a different mask family, producing a distinct corrupted image for the same clean target.

head and a ResNet-18 visual encoder. Per-task specialists ($\approx 85$M parameters) are paired with a wider generalist ($\approx 293$M parameters), realizing the theoretical capacity gap $\mathcal{C}_{\mathcal{F}} \gg \mathcal{C}_{\mathcal{H}}$. Specialists are trained on $\sim 900$ (StackCube) or $100$ (PickCube) labeled demonstrations per variant, then perform closed-loop pseudo-rollouts on unlabeled initial states. We apply success-checker rejection sampling to retain up to 800 successful trajectories per variant. The generalist is trained jointly on labeled + filtered pseudo data, while the multi-task baseline is trained on the labeled data only.

Table 1 reports closed-loop success rates over 100 rollouts per level. The semi-supervised MOL method improves over the labeled-only baseline in every setting across both task families. The gains are largest under the distribution shift. On StackCube, success improves by +16 percentage points at Level 2 and +14 points at Level 3; on PickCube, success improves by +44 points at Level 0 and +16 points at Level 3. These results suggest that the pseudo-rollouts provide useful additional supervision for training the larger generalist, particularly when the labeled demonstrations alone do not sufficiently cover the states encountered at test time.

## 5.2. Image Restoration

We instantiate the same recipe on CelebA-HQ (Karras et al., 2017) face inpainting at $64\times64$ with three mask families (Lugmayr et al., 2022) as the multiple objectives: alternating lines, face-region, and wide-stroke (Figure 3). The backbone is a Residual Denoising Diffusion Model (RDDM) (Liu et al., 2024), which decouples the reverse process into

Table 2. Performance comparison on CelebA-HQ inpainting at $64\times64$. We report PSNR, SSIM, and LPIPS on the 2k-image test split across three mask families.

| Mask | Semi-supervised MOL (ours) | | | Labeled-only MOL baseline | | | |
|---|---|---|---|---|---|---|---|
| | PSNR↑ | SSIM↑ | LPIPS↓ | PSNR↑ | SSIM↑ | LPIPS↓ | ΔPSNR |
| Alt. Lines | **24.351** | **0.817** | **0.058** | 22.939 | 0.763 | 0.072 | $+1.41$ |
| Face | **22.340** | **0.763** | **0.075** | 21.557 | 0.723 | 0.082 | $+0.78$ |
| Wide | **21.610** | **0.754** | **0.078** | 20.988 | 0.714 | 0.084 | $+0.62$ |

separate residual and noise prediction heads and is sampled with 10 steps. Each specialist has approximately 290M parameters, while the generalist has approximately 1.16B parameters. This gives a $4\times$ parameter increase and again realizes the intended capacity gap $C_{\mathcal{F}} \gg C_{\mathcal{H}}$.

The 28k-image training set is partitioned into 10k labeled images and 18k unlabeled images. For each mask family, we train one specialist on the labeled split and then use it to generate one pseudo-target for each unlabeled image, yielding 18k pseudo-pairs per objective. Unlike the robotics experiment, we do not apply rejection filtering here: all pseudo-pairs are retained. The ground-truth targets of the unlabeled split are never accessed during training.

Table 2 reports Peak Signal-to-Noise Ratio (PSNR), Structural Similarity Index Measure (SSIM), and Learned Perceptual Image Patch Similarity (LPIPS) (Zhang et al., 2018) on the full 2k test split with deterministic mask sampling. Our approach improves over the labeled-only baseline on *every* mask family and *every* metric: PSNR by $+0.62$ to $+1.41\,\mathrm{dB}$, SSIM by $+0.040$ to $+0.054$, and LPIPS by $-0.006$ to $-0.014$. These results demonstrate that our framework effectively leverages unlabeled data to improve reconstruction quality in inpainting.

### 5.3. Discussion

Across robotic control and image restoration, the proposed semi-supervised MOL procedure consistently improves over the labeled-only MOL baseline. These two settings differ substantially in modality, evaluation protocol, and labeled-data scale, yet they exhibit the same qualitative trend: training the large generalist with specialist-generated pseudo-data is more effective than training the same generalist on the labeled split alone.

This trend is consistent with the main statistical message of our theory. When the generalist class is much larger than the specialist classes ($C_{\mathcal{F}} \gg C_{\mathcal{H}}$), directly fitting the generalist from limited paired data can be sample-inefficient. The two-stage procedure mitigates this bottleneck by learning specialists at a smaller labeled-sample cost and then using abundant unlabeled conditions to generate additional training data for the generalist. Beyond the in-distribution improvements, the robotics results also show especially large gains on the held-out domain-randomization levels, suggest-

ing that specialist-generated pseudo-rollouts can improve closed-loop performance under distribution shift. Overall, the experiments provide qualitative support for the proposed specialist-to-generalist mechanism.

## 6. Conclusion and Discussion

We introduce a principled framework for multi-objective learning (MOL) with conditional diffusion models in a semi-supervised regime, where paired samples are scarce but condition data are abundant. We propose a two-stage training pipeline that bootstraps a generalist model from lightweight specialists through pseudo-sampling. We establish nonasymptotic generalization guarantees in score matching loss and distribution estimation error, highlighting that the labeled-sample complexity depends on the specialists' complexity rather than the capacity of the generalist class, and we further sharpened the rates for linear scalarizations. We also extend the theory to diffusion policies in sequential decision making, accounting for distribution shift induced by on-policy rollouts. Experiments on robotic manipulation and image restoration show consistent gains over labeled-only multi-task training, providing qualitative support for the proposed specialist-to-generalist mechanism. More broadly, we hope this work advances the statistical understanding of MOL for diffusion models and motivates more sample-efficient training algorithms in practice.

Although this work takes the first step towards the statistical properties of semi-supervised learning for diffusion models with multiple objectives, there are several interesting directions that we plan to address in the future research. First, our guarantees focus on the $K$ target objectives and do not address out-of-distribution (OOD) generalization; yet robustness to unseen tasks or environments is often critical in applications such as robotics. Second, while we study MOL in diffusion models, the underlying semi-supervised MOL framework is more general and may extend to other generative paradigms, including large language models. Finally, on the technical side, our analysis relies on a realizability assumption for the specialist classes, which may be stringent in practice; developing guarantees under model misspecification is an important direction for future work.

## Impact Statement

This paper presents work whose goal is to advance the field of Machine Learning. There are many potential societal consequences of our work, none of which we feel must be specifically highlighted here.

## Acknowledgements

Y.H. and S.S. were supported by the U.S. Army Research Laboratory and the U.S. Army Research Office under Grant W911NF2010219, Office of Naval Research, and NSF. This work used Jetstream2 at Indiana University through allocation CIS240832 and CIS251212 from the Advanced Cyberinfrastructure Coordination Ecosystem: Services & Support (ACCESS) program, which is supported by National Science Foundation grants #2138259, #2138286, #2138307, #2137603, and #2138296.

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

# A. Experiment Details

This appendix collects the implementation details deferred from Section 5.

## A.1. Robotic Manipulation

**Hardware and observation/action space.**   We use a Franka Panda arm in front of a single front-facing $256\times256$ RGB camera. Observations and actions are both 9-dimensional joint-position vectors.

**Domain randomization.**   We use three types of domain randomization. Scene and material randomization changes the table, ground, and wall materials using curated subsets of ARNOLD and NVIDIA vMaterials. Lighting randomization uses either a distant light with randomized polar angle or a randomly-sized cylinder-light array at a fixed height, with randomized intensity and color temperature. Camera-pose randomization moves a subset of cameras to side-facing viewpoints. For each domain-randomization level $L \geq 1$, all enabled randomization factors are sampled jointly at the beginning of each episode.

**Models.**   All policies are CNN-based diffusion policies (Chi et al., 2025) with a DDPM (Ho et al., 2020) conditional U-Net1D action head and a ResNet-18 visual encoder. Receding-horizon rollout uses $n_{\text{obs}}{=}3$ observation steps and $n_{\text{act}}{=}4$ action steps per inference, for a closed-loop horizon of 8. Specialists use a lightweight U-Net (downsampling channels $[256, 512, 1024]$, $\approx 85$M parameters); the generalist doubles the U-Net width ($[512, 1024, 2048]$, $\approx 293$M parameters).

**Data and pseudo-rollout protocol.**   Expert demonstrations are collected by motion-planning rollouts in Isaac Sim. For each training variant, we start from a pool of 1000 expert trajectories. To simulate the limited-label regime, we use only a small labeled subset for specialist training: approximately 900 trajectories per variant for `StackCube` and 100 trajectories per variant for `PickCube`. The last 100 trajectories are held out for evaluation and are never used for training. Each specialist then performs closed-loop pseudo-rollouts from 5000 randomly sampled initial states under a fixed randomization seed. We apply rejection sampling using the simulator's binary success checker and retain up to 800 successful pseudo-trajectories per variant. The generalist is trained jointly on the union of expert and filtered pseudo data across all four training variants. The multi-task baseline is identical except that it is trained only on the labeled shards, without specialists or pseudo data.

**Optimization.**   All models are trained with AdamW, learning rate $10^{-4}$, batch size 32, seed 42. The two methods are matched at $\approx 225$k gradient steps so the comparison is at equal compute.

**Evaluation.**   For each task family, each method, and each DR level $\in \{0, 1, 2, 3\}$ we roll out 100 closed-loop trajectories from the held-out initial-state seeds. The metric is success rate over the rollout horizon, scored by the simulator's per-task success checker. The same final checkpoint is used for our method and for the baseline in every cell.

## A.2. Image Restoration

**Backbone.**   We use the Residual Denoising Diffusion Model (RDDM) with the `pred_res_noise` objective and a two-branch `UnetRes` architecture: one residual U-Net and one noise U-Net, with `dim_mults=(1,2,4,8)`. RDDM augments the standard diffusion trajectory with a deterministic residual stream,

$$x_t = x_0 + \bar{\alpha}_t(x_{\text{cond}} - x_0) + \bar{\beta}_t\epsilon, \qquad \epsilon \sim \mathcal{N}(0, I),$$

where $x_{\text{cond}}$ is the masked-image condition. The U-Net takes the noisy state $x_t$ as input and predicts both a residual $\hat{r}$ and noise $\hat{\epsilon}$, from which the clean image is reconstructed as

$$\hat{x}_0 = x_t - \bar{\alpha}_t\hat{r} - \bar{\beta}_t\hat{\epsilon}.$$

Reverse sampling is initialized at $x_T = x_{\text{cond}} + \epsilon$, so the condition enters through the endpoint of the reverse trajectory rather than by channel concatenation at every denoising step. Specialists use base channel width `dim=64`, with approximately 290M parameters across the two U-Net branches. The generalist uses `dim=128`, with approximately 1.16B parameters.

**Data split.**   The 28k-image CelebA-HQ training set is partitioned with a fixed seed into a labeled split $\mathcal{L}$ of 10k images and an unlabeled pool $\mathcal{U}$ of 18k images. We use a separate 2k-image test split for evaluation.

**Specialist training.** For each mask family, we train one specialist on the labeled split $\mathcal{L}$ using random masks drawn from that family. Each specialist is trained for 20k optimizer steps with batch size 64, gradient accumulation 2, effective batch size 128, learning rate $2 \times 10^{-4}$, and Adam.

**Pseudo-sample generation.** Each specialist generates one pseudo-target for every image in $\mathcal{U}$, producing 18k pseudo-pairs per mask family. We retain all pseudo-pairs without rejection filtering. Therefore, the ground-truth targets from the unlabeled split are never accessed during training.

**Generalist training and baseline.** The generalist is trained from scratch on the union of the three augmented per-task pools under equal-weight linear scalarization. We use batch size 64, gradient accumulation 2, effective batch size 128, learning rate $2 \times 10^{-4}$, Adam, and the same random seed as the specialists. The matched-budget baseline uses the same training pipeline, but samples uniformly across the three mask families from the labeled split only. Thus, the only difference between the two methods is the training data: our method uses 10k labeled pairs plus 18k pseudo-pairs per task, whereas the baseline uses only 10k labeled pairs per task.

**Checkpoint and evaluation protocol.** We report both methods at 20k optimizer steps. We evaluate on the full 2k-image CelebA-HQ test split for all three mask families. Sampling is deterministic, using 10 reverse-process steps and a fixed sequential mask order. We report PSNR, SSIM, and LPIPS with an AlexNet trunk, evaluating both methods at the same checkpoint.

## B. Proofs

### B.1. Preliminaries

**Lemma B.1.** *If $x_0 \sim p(x_0|y)$, the density of forward process $p_t(x|y)$ can be written as*

$$p_t(x|y) = \int \phi_t(x|x_0)p(x_0|y)\mathrm{d}x_0, \quad \phi_t(x|x_0) = \frac{1}{(2\pi\sigma_t^2)^{\frac{d_x}{2}}} \exp\left(-\frac{\|x - \alpha_t x_0\|^2}{2\sigma_t^2}\right). \tag{B.1}$$

*Besides, the score function has the form of*

$$\nabla_x \log p_t(x|y) = \int \nabla_x \log \phi_t(x|x_0)\frac{\phi_t(x|x_0)p(x_0|y)}{\int \phi_t(x|z)p(z|y)\mathrm{d}z}\mathrm{d}x_0 \tag{B.2}$$

$$= \frac{1}{\alpha_t}\int \nabla_x \log p(x_0|y)\frac{\phi_t(x|x_0)p(x_0|y)}{\int \phi_t(x|z)p(z|y)\mathrm{d}z}\mathrm{d}x_0. \tag{B.3}$$

*Proof.* (B.1) can be directly implied by the definition of forward process. And it yields

$$\nabla_x \log p_t(x|y) = \frac{\nabla_x p_t(x|y)}{p_t(x|y)}$$

$$= \frac{\int \nabla_x \phi_t(x|x_0)p(x_0|y)\mathrm{d}x_0}{\int \phi_t(x|x_0)p(x_0|y)\mathrm{d}x_0} \tag{B.4}$$

$$= \int \nabla_x \log \phi_t(x|x_0)\frac{\phi_t(x|x_0)p(x_0|y)}{\int \phi_t(x|z)p(z|y)\mathrm{d}z}\mathrm{d}x_0,$$

which is (B.2). Moreover, noticing that $\nabla_x \phi_t(x|x_0) = -\frac{1}{\alpha_t}\nabla_{x_0}\phi_t(x|x_0)$, then by integration by parts,

$$\frac{\int \nabla_x \phi_t(x|x_0)p(x_0|y)\mathrm{d}x_0}{\int \phi_t(x|x_0)p(x_0|y)\mathrm{d}x_0} = -\frac{1}{\alpha_t}\frac{\int \nabla_{x_0}\phi_t(x|x_0)p(x_0|y)\mathrm{d}x_0}{\int \phi_t(x|x_0)p(x_0|y)\mathrm{d}x_0}$$

$$= \frac{1}{\alpha_t}\frac{\int \phi_t(x|x_0)\nabla_{x_0}p(x_0|y)\mathrm{d}x_0}{\int \phi_t(x|x_0)p(x_0|y)\mathrm{d}x_0} \tag{B.5}$$

$$= \frac{1}{\alpha_t}\int \nabla_x \log p(x_0|y)\frac{\phi_t(x|x_0)p(x_0|y)}{\int \phi_t(x|z)p(z|y)\mathrm{d}z}\mathrm{d}x_0.$$

Hence (B.3) is proved. $\square$

**Lemma B.2** (Cheng et al. (2025), Lemma B.11). *Let $\Phi$ be a class of functions on domain $\Omega$ and $\mathbb{P}$ be a probability distribution over $\Omega$. Suppose that for any $\varphi \in \Phi$, $\|\varphi\|_{L^\infty(\Omega)} \leq b$, $\mathbb{E}_\mathbb{P}[\varphi] \geq 0$, and $\mathbb{E}_\mathbb{P}[\varphi^2] \leq B\mathbb{E}_\mathbb{P}[\varphi] + B_0$ for some $b, B, B_0 \geq 0$. Let $x_1, \cdots, x_n \overset{\text{i.i.d.}}{\sim} \mathbb{P}$ and $\phi_n$ be a positive, non-decreasing and sub-root function such that*

$$\mathcal{R}_n(\Phi_r) := \mathbb{E}_{\boldsymbol{\sigma}} \sup_{\varphi \in \Phi_r} \left| \frac{1}{n} \sum_{i=1}^n \sigma_i \varphi(x_i) \right| \leq \phi_n(r). \tag{B.6}$$

*where $\Phi_r := \left\{ \varphi \in \Phi : \frac{1}{n} \sum_{i=1}^n (\varphi(x_i))^2 \leq r \right\}$. Define the largest fixed point of $\phi_n$ as $r_n^*$. Then for some absolute constant $C'$, with probability no less than $1 - \delta$, it holds that for any $\varphi \in \Phi$,*

$$\mathbb{E}_\mathbb{P}[\varphi] \leq \frac{2}{n} \sum_{i=1}^n \varphi(x_i) + C'(B \vee b) \left( r_n^* + \frac{\log\left((\log n)/\delta\right)}{n} \right) + C'\sqrt{\frac{B_0 \log\left((\log n)/\delta\right)}{n}}, \tag{B.7}$$

$$\frac{1}{n} \sum_{i=1}^n \varphi(x_i) \leq 2\mathbb{E}_\mathbb{P}[\varphi] + C'(B \vee b) \left( r_n^* + \frac{\log\left((\log n)/\delta\right)}{n} \right) + C'\sqrt{\frac{B_0 \log\left((\log n)/\delta\right)}{n}}. \tag{B.8}$$

*Proof.* We follow the procedures in Bousquet (2002). Let $\epsilon_j = b2^{-j}$ and consider a sequence of classes

$$\Phi^{(j)} := \{\varphi \in \Phi : \epsilon_{j+1} < \mathbb{E}_\mathbb{P}[\varphi] \leq \epsilon_j\}. \tag{B.9}$$

Note that $\Phi = \cup_{j \geq 0} \Phi^{(j)}$ and for $\varphi \in \Phi^{(j)}$, $\mathbb{E}_\mathbb{P}[\varphi^2] \leq B\epsilon_k + B_0$. Let $j_0 = \lfloor \log_2 n \rfloor$. Then by Bousquet (2002, Lemma 6.1), it holds that with probability no less than $1 - \delta$, for any $j \leq j_0$ and $\varphi \in \Phi^{(j)}$,

$$\left| \frac{1}{n} \sum_{i=1}^n \varphi(x_i) - \mathbb{E}_\mathbb{P}[\varphi] \right| \lesssim \mathcal{R}_n(\Phi^{(j)}) + \sqrt{\frac{(B\epsilon_j + B_0) \log\left(\log(b/\epsilon_j)/\delta\right)}{n}} + \frac{b \log\left(\log(b/\epsilon_j)/\delta\right)}{n}, \tag{B.10}$$

$$\left| \frac{1}{n} \sum_{i=1}^n (\varphi(x_i))^2 - \mathbb{E}_\mathbb{P}[\varphi^2] \right| \lesssim b\mathcal{R}_n(\Phi^{(j)}) + \sqrt{\frac{b^2(B\epsilon_j + B_0) \log\left(\log(b/\epsilon_j)/\delta\right)}{n}} + \frac{b^2 \log\left(\log(b/\epsilon_j)/\delta\right)}{n}. \tag{B.11}$$

Besides, for $\varphi \in \cup_{k > k_0} \Phi^{(j)} =: \Phi^{(j_0:)}$,

$$\left| \frac{1}{n} \sum_{i=1}^n \varphi(x_i) - \mathbb{E}_\mathbb{P}[\varphi] \right| \lesssim \mathcal{R}_n(\Phi^{(j_0:)}) + \sqrt{\frac{(B\epsilon_{j_0} + B_0) \log\left(\log(n)/\delta\right)}{n}} + \frac{b \log\left((\log n)/\delta\right)}{n} \tag{B.12}$$

From now on we reason on the conjunction of (B.10), (B.11) and (B.12). Define

$$U_j = B\epsilon_j + B_0 + b\mathcal{R}_n(\Phi^{(k)}) + \sqrt{\frac{b^2(B\epsilon_j + B_0) \log\left(\log(b/\epsilon_j)/\delta\right)}{n}} + \frac{b^2 \log\left(\log(b/\epsilon_j)/\delta\right)}{n}. \tag{B.13}$$

and thus for any $\varphi \in \Phi^{(j)}$, we have $\frac{1}{n} \sum_{i=1}^n (\varphi(x_i))^2 \leq CU_j$ for some absolute constant $C$ by (B.11), indicating that $\mathcal{R}_n(\Phi^{(j)}) \leq \phi_n(CU_j) \leq \sqrt{C}\phi_n(U_j)$. For any $j \leq j_0$,

$$U_j \leq 2(B\epsilon_j + B_0) + b\sqrt{C}\phi_n(U_j) + \frac{2b^2 \log\left((\log n)/\delta\right)}{n}. \tag{B.14}$$

Since $\phi_n$ is non-decreasing and sub-root, the inequality above implies that

$$U_j \lesssim b^2 r_n^* + B\epsilon_j + B_0 + \frac{b^2 \log\left((\log n)/\delta\right)}{n} =: r_n(\epsilon_j). \tag{B.15}$$

Therefore, for any $\varphi \in \Phi^{(j)}, j \leq j_0$, by (B.10),

$$\left| \frac{1}{n} \sum_{i=1}^n \varphi(x_i) - \mathbb{E}_\mathbb{P}[\varphi] \right| \lesssim \phi_n(r_n(\epsilon_j)) + \sqrt{\frac{(B\epsilon_j + B_0) \log\left((\log n)/\delta\right)}{n}} + \frac{b \log\left((\log n)/\delta\right)}{n} \tag{B.16}$$

$$=: F_n(\epsilon_j).$$

Noticing that $\mathbb{E}_{\mathbb{P}}[\varphi] \leq \epsilon_j \leq 2\mathbb{E}_{\mathbb{P}}[\varphi]$, it reduces to

$$\left| \frac{1}{n} \sum_{i=1}^{n} \varphi(x_i) - \mathbb{E}_{\mathbb{P}}[\varphi] \right| \lesssim F_n(\mathbb{E}_{\mathbb{P}}[\varphi]). \tag{B.17}$$

Hence we have by noting that $F_n$ is also a non-decreasing sub-root function,

$$\mathbb{E}_{\mathbb{P}}[\varphi] \leq \frac{2}{n} \sum_{i=1}^{n} \varphi(x_i) + C'(B \vee b)\left( r_n^* + \frac{\log\left((\log n)/\delta\right)}{n} \right) + C'\sqrt{\frac{B_0 \log\left((\log n)/\delta\right)}{n}}, \tag{B.18}$$

$$\frac{1}{n} \sum_{i=1}^{n} \varphi(x_i) \leq 2\mathbb{E}_{\mathbb{P}}[\varphi] + C'(B \vee b)\left( r_n^* + \frac{\log\left((\log n)/\delta\right)}{n} \right) + C'\sqrt{\frac{B_0 \log\left((\log n)/\delta\right)}{n}}. \tag{B.19}$$

Here $C'$ is an absolute constant. Moreover, when $\varphi \in \Phi^{(j)}$ for $j > j_0$, we have $\mathbb{E}_{\mathbb{P}}[\varphi] \leq \frac{b}{n}$, and according to (B.12),

$$\left| \frac{1}{n} \sum_{i=1}^{n} \varphi(x_i) - \mathbb{E}_{\mathbb{P}}[\varphi] \right| \lesssim F_n(\varepsilon_{j_0}). \tag{B.20}$$

Hence the same bounds apply, which completes the proof. $\qquad\square$

**Lemma B.3.** *Suppose for any $y$, $\mathrm{KL}(\mathbb{P}(\cdot|y)\|\mathcal{N}(0,I)) \leq C_{\mathrm{KL}}$ for some constant $C_{\mathrm{KL}}$. Then*

$$\mathbb{E}_{y \sim \mathbb{P}^Y} \mathrm{TV}(\mathbb{P}_{\widehat{h}}(\cdot|y), \mathbb{P}(\cdot|y)) \lesssim \sqrt{T_0} \log^{\frac{d_x+1}{2}}(1/T_0) + e^{-T} + \sqrt{L_{\mathbb{P}}(\widehat{h})}. \tag{B.21}$$

*Proof.* With a little abuse of notation, we will use $p_t(x_t|y)$ to denote the conditional density of $x_t|y$ under $p(x|y)$. Consider the following backward processes

$$d\widehat{x}_t = (\widehat{x}_t + 2\widehat{h}(\widehat{x}_t, y, T-t))dt + \sqrt{2}dW_t, \ \widehat{x}_0 \sim \mathcal{N}(0,I), 0 \leq t \leq T - T_0, \tag{B.22}$$

$$d\widetilde{x}_t = (\widetilde{x}_t + 2\widehat{h}(\widetilde{x}_t, y, T-t))dt + \sqrt{2}dW_t, \ \widetilde{x}_0 \sim \mathbb{P}_T(\cdot|y), 0 \leq t \leq T - T_0, \tag{B.23}$$

$$d\bar{x}_t = (\bar{x}_t + 2\nabla \log p_{T-t}(\bar{x}_t|y))dt + \sqrt{2}dW_t, \ \bar{x}_0 \sim \mathbb{P}_T(\cdot|y), 0 \leq t \leq T - T_0. \tag{B.24}$$

Denote the distribution of $\widetilde{x}_t|y$ as $\widetilde{\mathbb{P}}_{T-t}(\cdot|y)$. And note that $\bar{x}_t|y \sim \mathbb{P}_{T-t}(\cdot|y)$ according to classic reverse-time SDE results (Anderson, 1982). Then by Fu et al. (2024, Lemma D.5),

$$\mathrm{TV}(\mathbb{P}_{T_0}(\cdot|y), \mathbb{P}_0(\cdot|y)) \lesssim \sqrt{T_0} \log^{\frac{d_x+1}{2}}(1/T_0). \tag{B.25}$$

At the same time, we apply Data Processing inequality and Pinsker's inequality to get

$$\mathrm{TV}(\mathbb{P}_{\widehat{h}}(\cdot|y), \widetilde{\mathbb{P}}_{T_0}(\cdot|y)) \leq \mathrm{TV}(\mathbb{P}_T(\cdot|y), \mathcal{N}(0,I)) \lesssim \sqrt{\mathrm{KL}(\mathbb{P}_T(\cdot|y)\|\mathcal{N}(0,I))} \lesssim \sqrt{\mathrm{KL}(\mathbb{P}_0(\cdot|y)\|\mathcal{N}(0,I))}e^{-T}. \tag{B.26}$$

Again according to Pinsker's inequality and Girsanov's Theorem (Oko et al., 2023, Proposition D.1),

$$\begin{aligned}
\mathrm{TV}(\mathbb{P}_{T_0}(\cdot|y), \widetilde{\mathbb{P}}_{T_0}(\cdot|y)) &\lesssim \sqrt{\mathrm{KL}(\mathbb{P}_{T_0}(\cdot|y)\|\widetilde{\mathbb{P}}_{T_0}(\cdot|y))} \\
&\lesssim \sqrt{\mathbb{E}_{t,\bar{x}_t|y}\|\widehat{h}(\bar{x}_t, y, T-t) - \nabla \log p_{T-t}(\bar{x}_t|y)\|^2} \\
&= \sqrt{\mathbb{E}_{x \sim \mathbb{P}(\cdot|y)}[\ell(x, y, \widehat{h}) - \ell(x, y, s_{\mathbb{P}}^*)]}.
\end{aligned} \tag{B.27}$$

We complete the proof by combining three inequalities above and taking expectation over $y \sim \mathbb{P}^Y$. $\qquad\square$

**Lemma B.4.** *Let two probability distributions $\mathbb{P}, \mathbb{Q} \in \mathcal{P}(\mathbb{R}^d)$ satisfy $\mathrm{TV}(\mathbb{P}, \mathbb{Q}) \leq \varepsilon$. For a measurable set $\Omega \subset \mathbb{R}^d$, define the truncated distribution $\widetilde{\mathbb{P}}(A) = \mathbb{P}(A \cap \Omega)/\mathbb{P}(\Omega)$ for any measurable set $A \subset \mathbb{R}^d$. If $\mathbb{Q}(\Omega) \geq 1 - \delta$ and $\delta + \varepsilon \leq \frac{1}{2}$, then we have $\mathrm{TV}(\widetilde{\mathbb{P}}, \mathbb{Q}) \leq \delta + 2\varepsilon$.*

*Proof.* Since $\mathrm{TV}(\mathbb{P}, \mathbb{Q}) \leq \varepsilon$, we have $|\mathbb{P}(\Omega) - \mathbb{Q}(\Omega)| \leq \varepsilon$. Therefore,

$$\mathbb{P}(\Omega) \geq \mathbb{Q}(\Omega) - \varepsilon \geq 1 - \delta - \varepsilon \geq \frac{1}{2}.$$

Notice that $\mathrm{TV}(\widetilde{\mathbb{P}}, \mathbb{P}) = 1 - \mathbb{P}(\Omega)$ and by the triangle inequality,

$$\mathrm{TV}(\widetilde{\mathbb{P}}, \mathbb{Q}) \leq \mathrm{TV}(\widetilde{\mathbb{P}}, \mathbb{P}) + \mathrm{TV}(\mathbb{P}, \mathbb{Q}) \leq (\delta + \varepsilon) + \varepsilon = \delta + 2\varepsilon. \tag{B.28}$$

This concludes the proof. $\square$

## B.2. Proof in Sec. 3.1

**Lemma B.5.** *Assume* $n \gtrsim d_x M_1^{\frac{1}{2}} \log(d_x M_1 M_0 K/\delta) + M_0 + \frac{\log(1/T_0)}{T - T_0}$. *Under Asps. 3.1 to 3.3, it holds with probability no less than* $1 - \delta$ *that for any* $k \in [K]$,

$$L_{\mathbb{P}_k}(\widehat{h}_k) \lesssim \left(M_0^2 + d_x M_1^2 \log\left(\frac{nK}{\delta}\right)\right) \cdot \frac{\log \mathcal{N}(\mathcal{H}_k, \|\cdot\|_{L^\infty(\Omega_R)}, \frac{1}{n^2}) + \log(K/\delta)}{n}, \tag{B.29}$$

*where* $R \lesssim \log^{\frac{1}{2}}(nK/\delta)$ *and* $\Omega_R := [-R, R]^{d_x} \times [0, 1]^{d_y} \times [T_0, T]$.

*Proof.* For notation compactness, we will omit the task index $k$ in the following statements. Consider the truncated function class defined on $\mathbb{R}^{d_x} \times [0, 1]^{d_y}$,

$$\Phi := \{(x, y) \mapsto \widetilde{\ell}(x, y, h) := (\ell(x, y, h) - \ell(x, y, s^*)) \cdot \mathbb{1}_{\|x\|_\infty \leq R} : h \in \mathcal{H}\}, \tag{B.30}$$

where the truncation radius $R \geq 1$ will be specified later. Since $p(x|y)$ is sub-gaussian, it is easy to show that with probability no less than $1 - 2nd_x \exp(-C_2' R^2)$, it holds that $\|x_i\|_\infty \leq R$ for all $1 \leq i \leq n$. Hence by definition, the empirical minimizer also satisfies $\widehat{h} = \arg\min_{h \in \mathcal{H}} \frac{1}{n} \sum_{i=1}^n \widetilde{\ell}(x_i, y_i, h)$. Below we reason conditioned on this event and verify the conditions required in Lemma B.2.

**Step 1.** To bound the individual loss, when $\|x\|_\infty \leq R$,

$$\begin{aligned}
|\widetilde{\ell}(x, y, \cdot)| &\leq \sup_{h \in \mathcal{H}} \mathbb{E}_{t, x_t} \|h(x_t, y, t) - \nabla \log \phi_t(x_t|x)\|^2 \\
&\leq \sup_{h \in \mathcal{H}} 2[\mathbb{E}_{t, x_t} \|h(x_t, y, t)\|^2 + \mathbb{E}_{t, x_t} \|\nabla \log \phi_t(x_t|x)\|^2] \\
&\leq 4\mathbb{E}_{t, x_t}[M_0^2 + M_1^2 \|x_t\|^2] + 2\mathbb{E}_{t, \epsilon \sim \mathcal{N}(0, I)} \|\epsilon\|^2 / \sigma_t \\
&\leq 4(M_0^2 + M_1^2(\|x\|^2 + d_x)) + 4d_x\left(1 + \frac{\log(1/T_0)}{T - T_0}\right) \\
&\leq 8d_x\left(M_1^2 R^2 + \frac{\log(1/T_0)}{T - T_0}\right) + 8M_0^2 =: M.
\end{aligned} \tag{B.31}$$

Here we leverage the fact $\phi_t(x_t|x) = \mathcal{N}(\alpha_t x, \sigma_t^2 I)$ and Asp. 3.2 to bound $\|h(x_t, y, t)\|$. When $\|x\|_\infty > R$, $\widetilde{\ell}(x, y, h) = 0$. Therefore we have $|\widetilde{\ell}(x, y, h)| \leq M$.

**Step 2.** To bound the second order moment,

$$\begin{aligned}
&\mathbb{E}_{(x,y) \sim \mathbb{P}}\left[\mathbb{1}_{\|x\|_\infty \leq R} \left(\ell(x, y, h) - \ell(x, y, s^*)\right)^2\right] \\
&= \mathbb{E}_{(x,y) \sim \mathbb{P}}\left[\mathbb{1}_{\|x\|_\infty \leq R} \left(\mathbb{E}_{t, x_t|x} \|h(x_t, y, t) - \nabla \log \phi_t(x_t|x)\|^2 - \|s^*(x_t, y, t) - \nabla \log \phi_t(x_t|x)\|^2\right)^2\right] \\
&\leq \mathbb{E}_{(x,y) \sim \mathbb{P}}\left[\mathbb{1}_{\|x\|_\infty \leq R} \left(\mathbb{E}_{t, x_t|x} \|h(x_t, y, t) - s^*(x_t, y, t)\|^2\right) \right. \\
&\qquad\qquad \left. \cdot \left(\mathbb{E}_{t, x_t|x} \|h(x_t, y, t) + s^*(x_t, y, t) - 2\nabla \log \phi_t(x_t|x)\|^2\right)\right] \\
&\leq 4M\mathbb{E}_{(x,y) \sim \mathbb{P}}\left[\mathbb{1}_{\|x\|_\infty \leq R} \left(\mathbb{E}_{t, x_t|x} \|h(x_t, y, t) - s^*(x_t, y, t)\|^2\right)\right] \\
&\leq 4M\mathbb{E}_{(x,y) \sim \mathbb{P}} \left(\ell(x, y, h) - \ell(x, y, s^*)\right) \\
&\leq 4M\mathbb{E}_{(x,y) \sim \mathbb{P}}[\widetilde{\ell}(x, y, h)] + 4C_1' M^2 \exp(-C_2' R^2).
\end{aligned} \tag{B.32}$$

In the last inequality, we invoke Eq. (B.31) and

$$
\begin{aligned}
&\mathbb{E}_{(x,y)\sim\mathbb{P}}[(\ell(x,y,h) - \ell(x,y,s^*)) \cdot \mathbb{1}_{\|x\|_\infty > R}] \\
&\leq \mathbb{E}_{(x,y)\sim\mathbb{P}}\left[4(M_0^2 + M_1^2(\|x\|^2 + d_x)) + 4d_x(1 + \frac{\log(1/T_0)}{T - T_0})\right] \cdot \mathbb{1}_{\|x\|_\infty > R} \\
&\leq \mathbb{E}_{(x,y)\sim\mathbb{P}}(M_1^2\|x\|^2 + M) \cdot \mathbb{1}_{\|x\|_\infty > R} + M \\
&\leq \int_{\|x\|_\infty \geq R} C_1(M_1^2\|x\|^2 + M)\exp(-C_2\|x\|^2)\mathrm{d}x \\
&\leq C_1' M \exp(-C_2' R^2).
\end{aligned}
\tag{B.33}
$$

**Step 3.** To bound the local Rademacher complexity, note that for any $h_1, h_2 \in \mathcal{H}$,

$$
\left\| \frac{1}{\sqrt{n}} \sum_{i=1}^n \sigma_i \widetilde{\ell}(x_i, y_i, h_1) - \frac{1}{\sqrt{n}} \sum_{i=1}^n \sigma_i \widetilde{\ell}(x_i, y_i, h_2) \right\|_{\psi_2} \leq 4\|\widetilde{\ell}(\cdot, \cdot, h_1) - \widetilde{\ell}(\cdot, \cdot, h_2)\|_{L^2(\widehat{\mathbb{P}}_n)},
\tag{B.34}
$$

where $\widehat{\mathbb{P}}_n := \frac{1}{n}\sum_{i=1}^n \delta_{(x_i,y_i)}$. Define $\Phi_r := \{\varphi \in \Phi : \frac{1}{n}\sum_{i=1}^n \varphi(x_i, y_i)^2 \leq r\}$ and it is easy to show that $\mathbf{diam}\big(\Phi_r, \|\cdot\|_{L^2(\widehat{\mathbb{P}}_n)}\big) \leq 2\sqrt{r}$. By Dudley's bound (Van Handel, 2014; Wainwright, 2019), there exists an absolute constant $C_0$ such that for any $\theta > 0$,

$$
\mathcal{R}_n(\Phi_r) \leq C_0 \left( \theta + \int_\theta^{2\sqrt{r}} \sqrt{\frac{\log \mathcal{N}(\Phi_r, \|\cdot\|_{L^2(\widehat{\mathbb{P}}_n)}, \varepsilon)}{n}} \, \mathrm{d}\varepsilon \right).
\tag{B.35}
$$

Since $\|x_i\| \leq R$,

$$
\begin{aligned}
\frac{1}{n}\sum_{i=1}^n (\widetilde{\ell}(x_i, y_i, h_1) - \widetilde{\ell}(x_i, y_i, h_2))^2 &= \frac{1}{n}\sum_{i=1}^n (\ell(x_i, y_i, h_1) - \ell(x_i, y_i, h_2))^2 \\
&\leq \frac{1}{n}\sum_{i=1}^n \left[\mathbb{E}_{t,x_t|x_i}\|h_1 - h_2\|^2\right] \cdot \left[\mathbb{E}_{t,x_t|x_i}\|h_1 + h_2 - 2\nabla\log\phi_t\|^2\right] \\
&\leq \frac{4M}{n}\sum_{i=1}^n \mathbb{E}_{t,x_t|x_i}\|h_1(x_t, y_i, t) - h_2(x_t, y_i, t)\|^2.
\end{aligned}
\tag{B.36}
$$

Notice that $x_t|x_i \sim \mathcal{N}(\alpha_t x_i, \sigma_t^2 I)$, we have $\mathbb{P}(\|x_t\|_\infty \geq 2R) \leq d_x\mathbb{P}(|\mathcal{N}(0,1)| \leq R) \leq 2d_x\exp(-CR^2)$ for some absolute constant $C$. Therefore,

$$
\begin{aligned}
&\mathbb{E}_{t,x_t|x_i}\|h_1(x_t, y_i, t) - h_2(x_t, y_i, t)\|^2 \\
&\leq \mathbb{E}_{t,x_t|x_i}[\mathbb{1}_{\|x_t\|\leq 2R} \cdot \|h_1(x_t, y_i, t) - h_2(x_t, y_i, t)\|^2] + C_1' M \exp(-C_2' R^2) \\
&\leq \|h_1 - h_2\|_{L^\infty(\Omega_R)}^2 + C_1' M \exp(-C_2' R^2),
\end{aligned}
\tag{B.37}
$$

where $\Omega_R := [-2R, 2R]^{d_x} \times [0,1]^{d_y} \times [T_0, T]$. Plug in the bound above,

$$
\sqrt{\frac{1}{n}\sum_{i=1}^n (\widetilde{\ell}(x_i, y_i, h_1) - \widetilde{\ell}(x_i, y_i, h_2))^2} \leq 2M^{\frac{1}{2}}\|h_1 - h_2\|_{L^\infty(\Omega_R)} + 2C_1'M^{\frac{1}{2}}\exp(-C_2'R^2/2).
\tag{B.38}
$$

Hence for any $\varepsilon \geq 4C_1'M^{\frac{1}{2}}\exp(-C_2'R^2/2)$,

$$
\log\mathcal{N}(\Phi_r, \|\cdot\|_{L^2(\widehat{\mathbb{P}}_n)}, \varepsilon) \leq \log\mathcal{N}(\mathcal{H}, \|\cdot\|_{L^\infty(\Omega_R)}, \varepsilon/(4M^{\frac{1}{2}})).
\tag{B.39}
$$

Plug in Eq. (B.35) and let $\theta = \frac{4M^{\frac{1}{2}}}{n^2} \geq 4C_1' M^{\frac{1}{2}} \exp(-C_2' R^2/2)$,

$$
\begin{aligned}
\mathcal{R}_n(\Phi_r) &\leq C_0 \left( \frac{4M^{\frac{1}{2}}}{n^2} + \int_{4\sqrt{M}/n^2}^{2\sqrt{r}} \sqrt{\frac{\log \mathcal{N}(\mathcal{H}, \|\cdot\|_{L^\infty(\Omega_R)}, \varepsilon/(4M^{\frac{1}{2}}))}{n}} d\varepsilon \right) \\
&\leq C_0 \left( \frac{4M^{\frac{1}{2}}}{n^2} + 2\sqrt{r} \cdot \sqrt{\frac{\log \mathcal{N}(\mathcal{H}, \|\cdot\|_{L^\infty(\Omega_R)}, \frac{1}{n^2})}{n}} \right) \\
&=: \widetilde{\mathcal{R}}_n(r).
\end{aligned}
$$

(B.40)

Combine the steps above, by Lemma B.2 with $B_0 = 4C_1' M^2 \exp(-C_2' R^2)$, $B = 4M, b = M$, it holds with probability no less than $1 - 2nd_x \exp(-C_2' R^2) - \delta/2$, for any $h \in \mathcal{H}$,

$$
\begin{aligned}
\mathbb{E}_{(x,y)\sim\mathbb{P}}[\widetilde{\ell}(x,y,h)] &\leq \frac{2}{n} \sum_{i=1}^n \widetilde{\ell}(x_i, y_i, h) + C_3 M \left( r_n^* + \frac{\log(\log(n)/\delta)}{n} \right) \\
&\quad + C_3 \sqrt{\frac{M^2 \log(\log(n)/\delta)}{n}} \exp(-C_2' R^2/2),
\end{aligned}
$$

(B.41)

$$
\begin{aligned}
\frac{1}{n} \sum_{i=1}^n \widetilde{\ell}(x_i, y_i, h) &\leq 2\mathbb{E}_{(x,y)\sim\mathbb{P}}[\widetilde{\ell}(x,y,h)] + C_3 M \left( r_n^* + \frac{\log(\log(n)/\delta)}{n} \right) \\
&\quad + C_3 \sqrt{\frac{M^2 \log(\log(n)/\delta)}{n}} \exp(-C_2' R^2/2).
\end{aligned}
$$

(B.42)

where $r_n^*$ is the largest fixed point of $\widetilde{\mathcal{R}}_n$, and it can be bounded as

$$
r_n^* \leq C_4 \left( \frac{\sqrt{M}}{n^2} + \frac{\log \mathcal{N}(\mathcal{H}, \|\cdot\|_{L^\infty(\Omega_R)}, \frac{1}{n^2})}{n} \right),
$$

(B.43)

for some absolute constant $C_4$. Moreover, combine Eq. (B.33) with Eqs. (B.41) and (B.42),

$$
\begin{aligned}
\mathbb{E}_{(x,y)\sim\mathbb{P}}[\ell(x,y,h) - \ell(x,y,s^*)] &\leq \frac{2}{n} \sum_{i=1}^n [\ell(x_i, y_i, h) - \ell(x_i, y_i, s^*)] \\
&\quad + C_5 M \left( r_n^* + \frac{\log(\log(n)/\delta)}{n} + \exp(-C_2' R^2) \right),
\end{aligned}
$$

(B.44)

$$
\begin{aligned}
\frac{1}{n} \sum_{i=1}^n [\ell(x_i, y_i, h) - \ell(x_i, y_i, s^*)] &\leq 2\mathbb{E}_{(x,y)\sim\mathbb{P}}[\ell(x,y,h) - \ell(x,y,s^*)] \\
&\quad + C_5 M \left( r_n^* + \frac{\log(\log(n)/\delta)}{n} + \exp(-C_2' R^2) \right).
\end{aligned}
$$

(B.45)

Plug in the definition of $M = 8d_x(M_1^2 R^2 + \frac{\log(1/T_0)}{T-T_0}) + 8M_0^2$ and let $R = C \log^{\frac{1}{2}}(nd_x M_1 M_0/\delta)$ for some large constant $C$. Hence Eqs. (B.44) and (B.45) reduce to

$$
\begin{aligned}
\mathbb{E}_{(x,y)\sim\mathbb{P}}[\ell(x,y,h) - \ell(x,y,s^*)] &\leq \frac{2}{n} \sum_{i=1}^n [\ell(x_i, y_i, h) - \ell(x_i, y_i, s^*)] \\
&\quad + C_6 \left( M_0^2 + d_x M_1^2 \log(\frac{n}{\delta}) \right) \left( r_n^\dagger + \frac{\log(1/\delta)}{n} \right),
\end{aligned}
$$

(B.46)

$$
\begin{aligned}
\frac{1}{n} \sum_{i=1}^n [\ell(x_i, y_i, h) - \ell(x_i, y_i, s^*)] &\leq 2\mathbb{E}_{(x,y)\sim\mathbb{P}}[\ell(x,y,h) - \ell(x,y,s^*)] \\
&\quad + C_6 \left( M_0^2 + d_x M_1^2 \log(\frac{n}{\delta}) \right) \left( r_n^\dagger + \frac{\log(1/\delta)}{n} \right),
\end{aligned}
$$

(B.47)

where $r_n^\dagger := \frac{\log \mathcal{N}(\mathcal{H}, \|\cdot\|_{L^\infty(\Omega_R)}, \frac{1}{n^2})}{n}$ and $C_6$ is a constant. Here we use the condition $n \gtrsim d_x M_1^{\frac{1}{2}} \log(d_x M_1 M_0/\delta) + M_0 + \frac{\log(1/T_0)}{T-T_0}$ to simplify the notation.

Define $M_{n,\delta} = M_0^2 + d_x M_1^2 \log(\frac{n}{\delta})$. Therefore, we obtain that with probability no less than $1 - \delta$, the population loss of the empirical minimizer $\widehat{h}$ can be bounded by

$$
\begin{aligned}
\mathbb{E}_{(x,y)\sim\mathbb{P}}[\ell(x,y,\widehat{h}) - \ell(x,y,s^*)] &\leq \frac{2}{n}\sum_{i=1}^n[\ell(x_i,y_i,\widehat{h}) - \ell(x_i,y_i,s^*)] + C_6 M_{n,\delta}\left(r_n^\dagger + \frac{\log(1/\delta)}{n}\right) \\
&\leq \inf_{h\in\mathcal{H}}\frac{2}{n}\sum_{i=1}^n[\ell(x_i,y_i,h) - \ell(x_i,y_i,s^*)] + C_6 M_{n,\delta}\left(r_n^\dagger + \frac{\log(1/\delta)}{n}\right) \quad \text{(B.48)} \\
&\leq 3C_6 M_{n,\delta}\left(r_n^\dagger + \frac{\log(1/\delta)}{n}\right),
\end{aligned}
$$

We conclude the proof by replacing $\delta$ with $\delta/K$ and invoking all arguments above for each task $k \in [K]$. $\qquad\square$

**Theorem B.6.** *Let $\mathcal{E}_{\mathcal{S}}(f) := \mathcal{S}(L_{\mathbb{P}_1}(f), \cdots, L_{\mathbb{P}_K}(f))$ and assume $N, n \gtrsim d_x M_1^{\frac{1}{2}} \log(d_x M_1 M_0 K/\delta) + M_0$. Define the truncated distribution $\widetilde{\mathbb{P}}_{\widehat{h}_k}(\cdot|y) = \mathbb{P}_{\widehat{h}_k}(\cdot|y) \cdot \frac{\mathbb{1}_{B_R}}{\mathbb{P}_{\widehat{h}_k}(B_R|y)}$ denote the distribution of $\mathbb{P}_{\widehat{h}_k}$ restricted on $B_R = [-R,R]^{d_x}$ and suppose $\widetilde{x}_i^k \sim \widetilde{\mathbb{P}}_{\widehat{h}_k}(\cdot|\widetilde{y}_i^k)$. Then under Asps. 3.1 to 3.3, with probability no less than $1 - \delta$ and proper configurations of $T, T_0$, it holds that for any scalarization $\mathcal{S}$, $\mathcal{E}_{\mathcal{S}}(\widehat{f}_{\mathcal{S}}) - \inf_{f\in\mathcal{F}}\mathcal{E}_{\mathcal{S}}(f) \leq \mathcal{S}(\varepsilon_1, \cdots, \varepsilon_K)$, where*

$$
\varepsilon_k \lesssim \left(M_0^2 + d_x M_1^2 \log(\frac{NK}{\delta})\right) \cdot \left(\sqrt{\frac{\log \mathcal{N}(\mathcal{H}_k, \|\cdot\|_{L^\infty(\Omega_R)}, \frac{1}{n^2}) + \log(\frac{K}{\delta})}{n}} + \sqrt{\frac{\log \mathcal{N}(\mathcal{F}, \|\cdot\|_{L^\infty(\Omega_R)}, \frac{1}{N})}{N}}\right),
$$

(B.49)

*with $R \lesssim \log^{\frac{1}{2}}(NK/\delta)$ and $\Omega_R := [-R,R]^{d_x} \times [0,1]^{d_y} \times [T_0,T]$.*

*Proof.* Let $\widetilde{\mathcal{E}}_{\mathcal{S}}(f) := \mathcal{S}(\widetilde{L}_1(f), \cdots, \widetilde{L}_K(f))$ and $f_{\mathcal{S}}^* = \arg\min_{f\in\mathcal{F}}\mathcal{E}_{\mathcal{S}}(f)$. Notice that

$$
\begin{aligned}
\mathcal{E}_{\mathcal{S}}(\widehat{f}_{\mathcal{S}}) - \mathcal{E}_{\mathcal{S}}(f_{\mathcal{S}}^*) &\leq \mathcal{E}_{\mathcal{S}}(\widehat{f}_{\mathcal{S}}) - \widetilde{\mathcal{E}}_{\mathcal{S}}(\widehat{f}_{\mathcal{S}}) + \widetilde{\mathcal{E}}_{\mathcal{S}}(f_{\mathcal{S}}^*) - \mathcal{E}_{\mathcal{S}}(f_{\mathcal{S}}^*) \\
&\leq 2\sup_{f\in\mathcal{F}}|\mathcal{E}_{\mathcal{S}}(f) - \widetilde{\mathcal{E}}_{\mathcal{S}}(f)| \quad\quad\quad\quad\quad\quad\quad\quad\quad\quad\quad\quad \text{(B.50)} \\
&\leq 2\sup_{f\in\mathcal{F}}\mathcal{S}(|L_{\mathbb{P}_1} - \widetilde{L}_1|, \cdots, |L_{\mathbb{P}_K} - \widetilde{L}_K|).
\end{aligned}
$$

$$
\begin{aligned}
|L_{\mathbb{P}_k}(f) - \widetilde{L}_k(f)| &= |\mathbb{E}_{(x,y)\sim\mathbb{P}_k}[\ell(x,y,f) - \ell(x,y,s_k^*)] - \frac{1}{N}\sum_{i=1}^N[\ell(\widetilde{x}_i^k,\widetilde{y}_i^k,f) - \ell(\widetilde{x}_i^k,\widetilde{y}_i^k,\widehat{h}_k)]| \\
&\leq \underbrace{|\mathbb{E}_{(x,y)\sim\mathbb{P}_k}[\ell(x,y,\widehat{h}_k) - \ell(x,y,s_k^*)]|}_{\text{①}} \\
&\quad + \underbrace{\left|\mathbb{E}_{(x,y)\sim\mathbb{P}_k}[\ell(x,y,f) - \ell(x,y,\widehat{h}_k)] - \mathbb{E}_{y\sim\mathbb{P}_k^Y, x\sim\widetilde{\mathbb{P}}_{\widehat{h}_k}(\cdot|y)}[\ell(x,y,f) - \ell(x,y,\widehat{h}_k)]\right|}_{\text{②}} \quad \text{(B.51)} \\
&\quad + \underbrace{\left|\mathbb{E}_{y\sim\mathbb{P}_k^Y, x\sim\widetilde{\mathbb{P}}_{\widehat{h}_k}(\cdot|y)}[\ell(x,y,f) - \ell(x,y,\widehat{h}_k)] - \frac{1}{N}\sum_{i=1}^N[\ell(\widetilde{x}_i^k,\widetilde{y}_i^k,f) - \ell(\widetilde{x}_i^k,\widetilde{y}_i^k,\widehat{h}_k)]\right|}_{\text{③}}.
\end{aligned}
$$

For ②, by the same arguments in Eq. (B.31),

$$
|\ell(x,y,f) - \ell(x,y,\widehat{h}_k)| \leq 8M_0^2 + 8d_x(M_1^2\|x\|^2 + \frac{\log(1/T_0)}{T-T_0}). \quad \text{(B.52)}
$$

Let $M_R := 8M_0^2 + 8d_x(M_1^2 R^2 + \frac{\log(1/T_0)}{T - T_0})$.

$$
\begin{aligned}
② &\le M_R \mathbb{E}_{y \sim \mathbb{P}_k^Y} \mathrm{TV}(\mathbb{P}_k(\cdot|y), \widetilde{\mathbb{P}}_{\widehat{h}_k}(\cdot|y)) + \left| \mathbb{E}_{(x,y) \sim \mathbb{P}_k}[\ell(x, y, f) - \ell(x, y, \widehat{h}_k)] \cdot \mathbb{1}_{\|x\|_\infty > R} \right| \\
&\le M_R \mathbb{E}_{y \sim \mathbb{P}_k^Y} \mathrm{TV}(\mathbb{P}_k(\cdot|y), \mathbb{P}_{\widehat{h}_k}(\cdot|y)) + 2M_R \mathbb{E}_{y \sim \mathbb{P}_k^Y}[1 - \mathbb{P}_{\widehat{h}_k}(B_R|y)] + C_1' M_R \exp(-C_2' R^2) \\
&\le M_R \mathbb{E}_{y \sim \mathbb{P}_k^Y} \mathrm{TV}(\mathbb{P}_k(\cdot|y), \mathbb{P}_{\widehat{h}_k}(\cdot|y)) + 2C_1' M_R \exp(-C_2' R^2)
\end{aligned}
\tag{B.53}
$$

Here we invoke the same arguments in Eq. (B.33) in the second inequality and Lemma B.4.

For ③, consider the truncated function class defined on $\mathbb{R}^{d_x} \times [0,1]^{d_y}$,

$$
\Psi_k := \{(x, y) \mapsto \bar{\ell}(x, y, f) := (\ell(x, y, f) - \ell(x, y, \widehat{h}_k)) \cdot \mathbb{1}_{\|x\|_\infty \le R} : f \in \mathcal{F}\},
\tag{B.54}
$$

By similar arguments in Step 1. in Lemma 3.1, $|\bar{\ell}(x, y, f)| \le M_R$ for any $f \in \mathcal{F}$. And since $\widetilde{x}_i^k \sim \widetilde{\mathbb{P}}_{\widehat{h}_k}$ is truncated, we have $\|\widetilde{x}_i^k\|_\infty \le R$ for all $1 \le i \le N$. According to Wainwright (2019, Thm. 4.10), it holds that with probability at least $1 - \delta/(2K)$, for any $f \in \mathcal{F}$,

$$
\left| \mathbb{E}_{y \sim \mathbb{P}_k^Y, x \sim \mathbb{P}_{\widehat{h}_k}(\cdot|y)} \bar{\ell}(x, y, f) - \frac{1}{N} \sum_{i=1}^N \bar{\ell}(\widetilde{x}_i^k, \widetilde{y}_i^k, f) \right| \le 2\mathcal{R}_N(\Psi_k) + M_R \sqrt{\frac{2 \log(2K/\delta)}{N}}.
\tag{B.55}
$$

Note that for any $f_1, f_2 \in \mathcal{F}$,

$$
\left\| \frac{1}{\sqrt{N}} \sum_{i=1}^N \sigma_i \bar{\ell}(\widetilde{x}_i^k, \widetilde{y}_i^k, f_1) - \frac{1}{\sqrt{N}} \sum_{i=1}^N \sigma_i \bar{\ell}(\widetilde{x}_i^k, \widetilde{y}_i^k, f_2) \right\|_{\psi_2} \le 4\|\bar{\ell}(\cdot, \cdot, f_1) - \bar{\ell}(\cdot, \cdot, f_2)\|_{L^2(\widetilde{\mathbb{P}}_k)},
\tag{B.56}
$$

where $\widetilde{\mathbb{P}}_k := \frac{1}{N} \sum_{i=1}^N \delta_{(\widetilde{x}_i^k, \widetilde{y}_i^k)}$. It is easy to show that $\mathbf{diam}(\Psi_k, \|\cdot\|_{L^2(\widetilde{\mathbb{P}}_k)}) \le 2M_R$. By Dudley's bound (Van Handel, 2014; Wainwright, 2019), there exists an absolute constant $C_0$ such that for any $\theta > 0$,

$$
\mathcal{R}_N(\Psi_k) \le C_0 \left( \theta + \int_\theta^{2M_R} \sqrt{\frac{\log \mathcal{N}(\Psi_k, \|\cdot\|_{L^2(\widetilde{\mathbb{P}}_k)}, \varepsilon)}{N}} \, d\varepsilon \right).
\tag{B.57}
$$

By the same arguments in Step 3. in Lemma 3.1, for any $\varepsilon \ge 4C_1' M_R^{\frac{1}{2}} \exp(-C_2' R^2/2)$,

$$
\log \mathcal{N}(\Psi_k, \|\cdot\|_{L^2(\widetilde{\mathbb{P}}_k)}, \varepsilon) \le \log \mathcal{N}(\mathcal{F}, \|\cdot\|_{L^\infty(\Omega_R)}, \varepsilon/(4M_R^{\frac{1}{2}})).
\tag{B.58}
$$

Let $\theta = \frac{4M_R^{\frac{1}{2}}}{N} \ge 4C_1' M_R^{\frac{1}{2}} \exp(-C_2' R^2/2)$, and we achieve

$$
\begin{aligned}
\mathcal{R}_N(\Psi_k) &\le C_0 \left( \frac{4M_R^{\frac{1}{2}}}{N} + \int_{4\sqrt{M_R}/N}^{2M_R} \sqrt{\frac{\log \mathcal{N}(\mathcal{F}, \|\cdot\|_{L^\infty(\Omega_R)}, \varepsilon/(4M_R^{\frac{1}{2}}))}{N}} d\varepsilon \right) \\
&\le C_0 \left( \frac{4M_R^{\frac{1}{2}}}{N} + 2M_R \cdot \sqrt{\frac{\log \mathcal{N}(\mathcal{F}, \|\cdot\|_{L^\infty(\Omega_R)}, \frac{1}{N})}{N}} \right).
\end{aligned}
\tag{B.59}
$$

Therefore, with probability no less than $1 - \delta/(2K)$,

$$
\begin{aligned}
\left| \mathbb{E}_{y \sim \mathbb{P}_k^Y, x \sim \widetilde{\mathbb{P}}_{\widehat{h}_k}(\cdot|y)} \bar{\ell}(x, y, f) - \frac{1}{N} \sum_{i=1}^N \bar{\ell}(\widetilde{x}_i^k, \widetilde{y}_i^k, f) \right| &\le 2\mathcal{R}_N(\Psi_k) + M_R \sqrt{\frac{2 \log(2K/\delta)}{N}} \\
&\le C_3 M_R \sqrt{\frac{\log \mathcal{N}(\mathcal{F}, \|\cdot\|_{L^\infty(\Omega_R)}, \frac{1}{N}) + \log(2K/\delta)}{N}}.
\end{aligned}
\tag{B.60}
$$

Consequently,

$$③ \leq C_1' M_R \exp(-C_2' R^2) + C_3 M_R \sqrt{\frac{\log \mathcal{N}(\mathcal{F}, \|\cdot\|_{L^\infty(\Omega_R)}, \frac{1}{N}) + \log(K/\delta)}{N}}. \tag{B.61}$$

Let $R = C \log^{\frac{1}{2}}(NKM_1M_0d_x/\delta)$ for some large constant $C$ and then we have

$$M_R \leq C_4 \left( M_0^2 + d_x M_1^2 \log(\frac{NK}{\delta}) + \frac{\log(1/T_0)}{T - T_0} \right). \tag{B.62}$$

Here we use the condition that $N \gtrsim d_x M_1^{\frac{1}{2}} \log(d_x M_1 K/\delta) + M_0 + \frac{\log(1/T_0)}{T-T_0}$.

Finally, let $T = \mathcal{O}(\log n), T_0 = \mathcal{O}\left(1/(N^2 \log^{d_x+1}(N))\right)$. Then $M_R \lesssim M_0^2 + d_x M_1^2 \log(\frac{NK}{\delta})$ and with probability at least $1 - \delta$, for any $k \in [K]$ and $f \in \mathcal{F}$,

$$|L_{\mathbb{P}_k}(f) - \widetilde{L}_k(f)|$$
$$\leq ① + ② + ③$$

$$\lesssim L_{\mathbb{P}_k}(\widehat{h}_k) + M_R \mathbb{E}_{y \sim \mathbb{P}_k^Y} \text{TV}(\mathbb{P}_k(\cdot|y), \mathbb{P}_{\widehat{h}_k}(\cdot|y)) + M_R \sqrt{\frac{\log \mathcal{N}(\mathcal{F}, \|\cdot\|_{L^\infty(\Omega_R)}, \frac{1}{N}) + \log(K/\delta)}{N}}$$

$$\lesssim L_{\mathbb{P}_k}(\widehat{h}_k) + M_R(\frac{1}{n} + \sqrt{L_{\mathbb{P}_k}(\widehat{h}_k)}) + M_R \sqrt{\frac{\log \mathcal{N}(\mathcal{F}, \|\cdot\|_{L^\infty(\Omega_R)}, \frac{1}{N}) + \log(K/\delta)}{N}} \tag{B.63}$$

$$\lesssim \left( M_0^2 + d_x M_1^2 \log(\frac{NK}{\delta}) \right) \cdot \left( \sqrt{\frac{\log \mathcal{N}(\mathcal{H}_k, \|\cdot\|_{L^\infty(\Omega_R)}, \frac{1}{n^2}) + \log(\frac{K}{\delta})}{n}} + \sqrt{\frac{\log \mathcal{N}(\mathcal{F}, \|\cdot\|_{L^\infty(\Omega_R)}, \frac{1}{N})}{N}} \right)$$

Here in the third inequality we invoke Lemma B.3. We conclude the proof by plugging the bound above into Eq. (B.50). $\square$

**Corollary B.7.** *Suppose $\mathcal{S}(|\cdot|)$ is coordinatewise non-decreasing and $|\mathcal{S}(\vec{u})| \leq \|\vec{u}\|_\infty$. Under the same conditions in Thm. B.6,*

$$\mathcal{S}\left(\left(\mathbb{E}_{y \sim \mathbb{P}_k^Y} \text{TV}(\mathbb{P}_{\widehat{f}_\mathcal{S}}(\cdot|y), \mathbb{P}_k(\cdot|y))\right)_k\right) \lesssim \varepsilon_{apx}(\mathcal{S}) + \mathcal{S}^{\frac{1}{2}}(\varepsilon_1^2, \cdots, \varepsilon_K^2), \tag{B.64}$$

*where $\varepsilon_k$ is bounded by Eq. (3.2).*

*Proof.* We first claim that for any $\vec{u} \geq 0, \mathcal{S}(\vec{u}^2) \geq \mathcal{S}(\vec{u})^2$. In fact, by reverse-triangle inequality and positive homogeneity of $\mathcal{S}$, it is easy to show that for any $\vec{u} \in \mathbb{R}_+^K, \mathcal{S}(\vec{u}) \geq \mathcal{S}(0) = 0$. Define $F(\vec{u}) := \mathcal{S}(\vec{u}_+)$ where $\vec{u}_+ = \max\{\vec{u}, 0\}$. Then $F$ also satisfies reverse-triangle inequality and positive homogeneity. Let

$$C := \{\vec{v} \in \mathbb{R}^K : \vec{v} \cdot \vec{u} \leq F(\vec{u}), \forall \vec{u} \in \mathbb{R}^K\}. \tag{B.65}$$

According to Hahn–Banach separation (see e.g., Simons (2008, Coro. 2.4)),

$$F(\vec{u}) = \sup_{\vec{v} \in C} \vec{v} \cdot \vec{u}, \forall \vec{u} \in \mathbb{R}^K. \tag{B.66}$$

Further notice that $0 \leq F(\vec{u}) \leq \|\vec{u}\|_\infty$, hence for any $\vec{v} \in C, \vec{v} \geq 0$ and $\sum_{k=1}^K \vec{v}_k \leq 1$. By Cauchy–Schwarz inequality,

$$(\vec{v} \cdot \vec{u})^2 \leq (\sum_{k=1}^K \vec{v}_k) \cdot (\sum_{k=1}^K \vec{v}_k \vec{u}_k^2) \leq \sum_{k=1}^K \vec{v}_k \vec{u}_k^2 = \vec{v} \cdot \vec{u}^2. \tag{B.67}$$

Therefore for any $\vec{u} \geq 0, \mathcal{S}(\vec{u}^2) = F(\vec{u}^2) = \sup_{\vec{v} \in C} \vec{v} \cdot \vec{u}^2 \geq \sup_{\vec{v} \in C}(\vec{v} \cdot \vec{u})^2 = F(\vec{u})^2 = \mathcal{S}(\vec{u})^2$.

By Lemma B.3 and noticing that $T = \mathcal{O}(\log N), T_0 = \mathcal{O}\left(1/(N^2 \log^{d_x+1}(N))\right)$, we obtain

$$\mathcal{S}\left(\left(\mathbb{E}_{y \sim \mathbb{P}_k^Y} \text{TV}(\mathbb{P}_{\widehat{f}_\mathcal{S}}(\cdot|y), \mathbb{P}_k(\cdot|y))\right)_k\right) \leq \mathcal{S}\left(\left(\sqrt{L_{\mathbb{P}_k}(\widetilde{f}_\mathcal{S})}\right)_k\right) + \mathcal{O}(\frac{1}{N})$$
$$\leq \mathcal{S}^{\frac{1}{2}}\left(\left(L_{\mathbb{P}_k}(\widetilde{f}_\mathcal{S})\right)_k\right) + \mathcal{O}(\frac{1}{N}) \tag{B.68}$$
$$\lesssim \varepsilon_{\text{apx}}(\mathcal{S}) + \mathcal{S}^{\frac{1}{2}}(\varepsilon_1^2, \cdots, \varepsilon_K^2).$$

$\square$

## B.3. Proof in Sec. 3.2

**Theorem B.8.** *Under the same conditions of Thm. B.6, further assume that $\mathcal{F}$ is convex. Define the truncated distribution* $\widetilde{\mathbb{P}}_{\widehat{h}_k}(\cdot|y) = \mathbb{P}_{\widehat{h}_k}(\cdot|y) \cdot \frac{\mathbb{1}_{B_R}}{\mathbb{P}_{\widehat{h}_k}(B_R|y)}$ *denote the distribution of $\mathbb{P}_{\widehat{h}_k}$ restricted on $B_R = [-R, R]^{d_x}$ and suppose $\widetilde{x}_i^k \sim \widetilde{\mathbb{P}}_{\widehat{h}_k}(\cdot|\widetilde{y}_i^k)$.*
*Then the following holds with probability no less than $1 - \delta$ and proper configurations of $T, T_0$: for any $\mathcal{S} \in \mathcal{S}_{lin}$ with $\mathcal{S}(\vec{u}) = \sum_{k=1}^K \lambda_k \vec{u}_k$,*

$$\mathcal{S}\left(\left(\mathbb{E}_{y \sim \mathbb{P}_k^Y} \mathrm{TV}(\mathbb{P}_{\widehat{f}_{\mathcal{S}}}(\cdot|y), \mathbb{P}_k(\cdot|y))\right)_k\right) \lesssim \bar{\varepsilon}_{apx}(\mathcal{S})$$

$$+ \left(M_0 + M_1 d_x^{\frac{1}{2}} \log^{\frac{1}{2}}\left(\frac{NK}{\delta}\right)\right)\left(\sqrt{\frac{\log \mathcal{N}(\mathcal{F}, \|\cdot\|_{\Omega_R}, \frac{1}{N})}{N}} + \sum_{k=1}^K \lambda_k \sqrt{\frac{\log \mathcal{N}(\mathcal{H}_k, \|\cdot\|_{\Omega_R}, \frac{1}{n^2}) + \log(\frac{K}{\delta})}{n}}\right).$$
(B.69)

*Proof.* Let $\widetilde{f}_{\mathcal{S}} := \arg\min_{f \in \mathcal{F}} \sum_{k=1}^K \lambda_k \mathbb{E}_{y \sim \mathbb{P}_k^Y, x \sim \widehat{\mathbb{P}}_{\widehat{h}_k}(\cdot|y)}[\ell(x, y, f) - \ell(x, y, \widehat{h}_k)]$. For any $f \in \mathcal{F}$ and $\alpha \in [0, 1]$, we have $f_\alpha := \alpha f + (1 - \alpha)\widetilde{f}_{\mathcal{S}} \in \mathcal{F}$ due to the convexity of $\mathcal{F}$. By the definition of $\widetilde{f}_{\mathcal{S}}$,

$$0 \leq \sum_{k=1}^K \lambda_k \mathbb{E}_{y \sim \mathbb{P}_k^Y, x \sim \widetilde{\mathbb{P}}_{\widehat{h}_k}(\cdot|y)}[\ell(x, y, f_\alpha) - \ell(x, y, \widetilde{f}_{\mathcal{S}})]$$

$$= \sum_{k=1}^K \lambda_k \mathbb{E}_{y \sim \mathbb{P}_k^Y, x \sim \widetilde{\mathbb{P}}_{\widehat{h}_k}(\cdot|y)} \mathbb{E}_{t, x_t}[\|f_\alpha(x_t, y, t) - \nabla \log \phi_t(x_t|x)\|^2 - \|\widetilde{f}_{\mathcal{S}}(x_t, y, t) - \nabla \log \phi_t(x_t|x)\|^2] \qquad \text{(B.70)}$$

$$= \sum_{k=1}^K \lambda_k \mathbb{E}_{y \sim \mathbb{P}_k^Y, x \sim \widetilde{\mathbb{P}}_{\widehat{h}_k}(\cdot|y)} \mathbb{E}_{t, x_t}[(f_\alpha - \widetilde{f}_{\mathcal{S}})^\top (f_\alpha + \widetilde{f}_{\mathcal{S}} - 2\nabla \log \phi_t)].$$

Let $\alpha \to 0$ and it indicates that,

$$\sum_{k=1}^K \lambda_k \mathbb{E}_{y \sim \mathbb{P}_k^Y, x \sim \widetilde{\mathbb{P}}_{\widehat{h}_k}(\cdot|y)} \mathbb{E}_{t, x_t}[(f - \widetilde{f}_{\mathcal{S}})^\top (\widetilde{f}_{\mathcal{S}} - \nabla \log \phi_t)] \geq 0. \qquad \text{(B.71)}$$

Therefore for any $f \in \mathcal{F}$,

$$\sum_{k=1}^K \lambda_k \mathbb{E}_{y \sim \mathbb{P}_k^Y, x \sim \widetilde{\mathbb{P}}_{\widehat{h}_k}(\cdot|y)}[\ell(x, y, f) - \ell(x, y, \widetilde{f}_{\mathcal{S}})]$$

$$\geq \sum_{k=1}^K \lambda_k \mathbb{E}_{y \sim \mathbb{P}_k^Y, x \sim \widetilde{\mathbb{P}}_{\widehat{h}_k}(\cdot|y)} \mathbb{E}_{t, x_t} \|f(x_t, y, t) - \widetilde{f}_{\mathcal{S}}(x_t, y, t)\|^2. \qquad \text{(B.72)}$$

Let $\mathcal{F}_{\mathcal{S}, k}(r) := \{f \in \mathcal{F} : \mathbb{E}_{y \sim \mathbb{P}_k^Y, x \sim \widetilde{\mathbb{P}}_{\widehat{h}_k}(\cdot|y)} \mathbb{E}_{t, x_t} \|f(x_t, y, t) - \widetilde{f}_{\mathcal{S}}(x_t, y, t)\|^2 \leq r^2\}$ and $\mathcal{M}_k(r) := \{(\mathcal{S}, f) := \mathcal{S} \in \mathcal{S}_{\text{lin}}, f \in \mathcal{F}_{\mathcal{S}, k}(r)\}$. Consider the truncated family class

$$\Psi_k(r) := \{(x, y) \mapsto \bar{\ell}(x, y, f, \mathcal{S}) := (\ell(x, y, f) - \ell(x, y, \widetilde{f}_{\mathcal{S}})) \cdot \mathbb{1}_{\|x\|_\infty \leq R} : (\mathcal{S}, f) \in \mathcal{M}_k(r)\}. \qquad \text{(B.73)}$$

By definition, we have $\|\widetilde{x}_i^k\|_\infty \leq R$ for all $1 \leq i \leq N$.

**Step 1.** To bound the individual loss, by the same arguments in Eq. (B.31),

$$|\bar{\ell}(x, y, \cdot, \cdot)| \leq 8d_x \left(M_1^2 R^2 + \frac{\log(1/T_0)}{T - T_0}\right) + 8M_0^2 =: M_R. \qquad \text{(B.74)}$$

**Step 2.** To bound the second order moment, for any $(\mathcal{S}, f) \in \mathcal{M}_k(r)$,

$$\mathbb{E}_{y \sim \mathbb{P}_k^Y, x \sim \widetilde{\mathbb{P}}_{\hat{h}_k}(\cdot|y)} \left[ \mathbb{1}_{\|x\|_\infty \leq R} \left( \ell(x, y, f) - \ell(x, y, \widetilde{f}_{\mathcal{S}}) \right)^2 \right]$$

$$= \mathbb{E}_{y \sim \mathbb{P}_k^Y, x \sim \widetilde{\mathbb{P}}_{\hat{h}_k}(\cdot|y)} \left[ \mathbb{1}_{\|x\|_\infty \leq R} \left( \mathbb{E}_{t, x_t} \| f(x_t, y, t) - \nabla \log \phi_t(x_t|x) \|^2 - \| \widetilde{f}_{\mathcal{S}}(x_t, y, t) - \nabla \log \phi_t(x_t|x) \|^2 \right)^2 \right]$$

$$\leq \mathbb{E}_{y \sim \mathbb{P}_k^Y, x \sim \widetilde{\mathbb{P}}_{\hat{h}_k}(\cdot|y)} \left[ \mathbb{1}_{\|x\|_\infty \leq R} \left( \mathbb{E}_{t, x_t} \| f(x_t, y, t) - \widetilde{f}_{\mathcal{S}}(x_t, y, t) \|^2 \right) \right.$$
$$\left. \cdot \left( \mathbb{E}_{t, x_t} \| f(x_t, y, t) + \widetilde{f}_{\mathcal{S}}(x_t, y, t) - 2 \nabla \log \phi_t(x_t|x) \|^2 \right) \right]$$

$$\leq 4 M_R \mathbb{E}_{y \sim \mathbb{P}_k^Y, x \sim \widetilde{\mathbb{P}}_{\hat{h}_k}(\cdot|y)} \left[ \mathbb{1}_{\|x\|_\infty \leq R} \left( \mathbb{E}_{t, x_t} \| f(x_t, y, t) - \widetilde{f}_{\mathcal{S}}(x_t, y, t) \|^2 \right) \right]$$

$$\leq 4 M_R \mathbb{E}_{y \sim \mathbb{P}_k^Y, x \sim \widetilde{\mathbb{P}}_{\hat{h}_k}(\cdot|y)} \left[ \mathbb{E}_{t, x_t} \| f(x_t, y, t) - \widetilde{f}_{\mathcal{S}}(x_t, y, t) \|^2 \right]$$

$$\leq 4 M_R r^2. \tag{B.75}$$

**Step 3.** To bound the Rademacher complexity, note that for any $(\mathcal{S}_1, f_1), (\mathcal{S}_2, f_2) \in \mathcal{M}_k(r)$,

$$\left\| \frac{1}{\sqrt{N}} \sum_{i=1}^N \sigma_i \bar{\ell}(\widetilde{x}_i^k, \widetilde{y}_i^k, f_1, \mathcal{S}_1) - \frac{1}{\sqrt{N}} \sum_{i=1}^N \sigma_i \bar{\ell}(\widetilde{x}_i^k, \widetilde{y}_i^k, f_2, \mathcal{S}_2) \right\|_{\psi_2} \leq 4 \| \bar{\ell}(\cdot, \cdot, f_1, \mathcal{S}_1) - \bar{\ell}(\cdot, \cdot, f_2, \mathcal{S}_2) \|_{L^2(\widetilde{\mathbb{P}}_k)}, \tag{B.76}$$

where $\widetilde{\mathbb{P}}_k := \frac{1}{N} \sum_{i=1}^N \delta_{(\widetilde{x}_i^k, \widetilde{y}_i^k)}$. Define $\mathbf{diam}(\Psi_k(r), \| \cdot \|_{L^2(\widetilde{\mathbb{P}}_k)}) = D_r$. By Dudley's bound (Van Handel, 2014; Wainwright, 2019), there exists an absolute constant $C_0$ such that for any $\theta > 0$,

$$\mathcal{R}_N(\Psi_k(r)) \leq C_0 \left( \theta + \int_\theta^{D_r} \sqrt{\frac{\log \mathcal{N}(\Psi_k(r), \| \cdot \|_{L^2(\widetilde{\mathbb{P}}_k)}, \varepsilon)}{N}} \, d\varepsilon \right). \tag{B.77}$$

By the same arguments in Step 3. in Lemma 3.1, for any $(\mathcal{S}_1, f_1), (\mathcal{S}_2, f_2) \in \mathcal{M}_k(r)$,

$$\sqrt{\frac{1}{N} \sum_{i=1}^N (\bar{\ell}(\widetilde{x}_i^k, \widetilde{y}_i^k, f_1, \mathcal{S}_\infty) - \bar{\ell}(\widetilde{x}_i^k, \widetilde{y}_i^k, f_2, \mathcal{S}_2))^2} \tag{B.78}$$
$$\leq 2 M_R^{\frac{1}{2}} [\| f_1 - f_2 \|_{L^\infty(\Omega_R)} + \| \widetilde{f}_{\mathcal{S}_1} - \widetilde{f}_{\mathcal{S}_2} \|_{L^\infty(\Omega_R)}] + 4 C_1' M_R^{\frac{1}{2}} \exp(-C_2' R^2/2).$$

Let $\mathcal{F}' := \{\widetilde{f}_{\mathcal{S}} : \mathcal{S} \in \mathcal{S}_{\text{lin}}\} \subseteq \mathcal{F}$. Hence for any $\varepsilon \geq 8 C_1' M_R^{\frac{1}{2}} \exp(-C_2' R^2/2)$,

$$\log \mathcal{N}(\Psi_k(r), \| \cdot \|_{L^2(\widetilde{\mathbb{P}}_k)}, \varepsilon) \leq \log \mathcal{N}(\mathcal{F}, \| \cdot \|_{L^\infty(\Omega_R)}, \varepsilon/(8 M_R^{\frac{1}{2}})) + \log \mathcal{N}(\mathcal{F}', \| \cdot \|_{L^\infty(\Omega_R)}, \varepsilon/(8 M_R^{\frac{1}{2}}))$$
$$\leq 2 \log \mathcal{N}(\mathcal{F}, \| \cdot \|_{L^\infty(\Omega_R)}, \varepsilon/(16 M_R^{\frac{1}{2}})). \tag{B.79}$$

Plug in the bound above and let $\theta = \frac{16 M_R^{\frac{1}{2}}}{N} \geq 8 C_1' M_R^{\frac{1}{2}} \exp(-C_2' R^2/2)$,

$$\mathcal{R}_N(\Psi_k(r)) \leq C_0 \left( \frac{16 M_R^{\frac{1}{2}}}{N} + \int_{16\sqrt{M_R}/N}^{D_r} \sqrt{\frac{\log \mathcal{N}(\Psi_k(r), \| \cdot \|_{L^2(\widetilde{\mathbb{P}}_k)}, \varepsilon)}{N}} \, d\varepsilon \right)$$
$$\leq C_0 \left( \frac{16 M_R^{\frac{1}{2}}}{N} + D_r \sqrt{\frac{\log \mathcal{N}(\mathcal{F}, \| \cdot \|_{\Omega_R}, \frac{1}{N})}{N}} \right). \tag{B.80}$$

Now we turn to bound $D_r$. Denote $r_j := 2^{-j} M_R^{\frac{1}{2}}$ and $j_0 := \lceil \log_2 N \rceil$. According to Bousquet (2002, Lemma 6.2),

the following holds with probability at least $1 - \delta/(4K)$: for any $0 \leq j \leq j_0, \psi \in \Psi_k(r_j)$,

$$
\left| \frac{1}{N} \sum_{i=1}^{N} \psi(\widetilde{x}_i^k, \widetilde{y}_i^k)^2 - \mathbb{E}_{y \sim \mathbb{P}_k^Y, x \sim \widetilde{\mathbb{P}}_{\widehat{h}_k}(\cdot|y)}[\psi(x,y)^2] \right|
$$
$$
\lesssim M_R \mathcal{R}_N(\Psi_k(r_j)) + M_R \sqrt{\frac{M_R r_j^2 \log(K \log(N)/\delta)}{N}} + \frac{M_R^2 \log(K \log(N)/\delta)}{N}. \tag{B.81}
$$

Combining this with Step 2., we get

$$
\left| \frac{1}{N} \sum_{i=1}^{N} \psi(\widetilde{x}_i^k, \widetilde{y}_i^k)^2 \right| \lesssim M_R r_j^2 + M_R \mathcal{R}_N(\Psi_k(r_j)) + \frac{M_R^2 \log(K \log(N)/\delta)}{N}. \tag{B.82}
$$

indicating that

$$
D_{r_j}^2 \lesssim M_R r_j^2 + M_R \mathcal{R}_N(\Psi_k(r_j)) + \frac{M_R^2 \log(K \log(N)/\delta)}{N}. \tag{B.83}
$$

Plug into Eq. (B.80), we finally get with probability no less than $1 - \delta/(4K)$, for any $j \leq j_0$,

$$
\mathcal{R}_N(\Psi_k(r_j)) \lesssim \frac{M_R \left( \log \mathcal{N}(\mathcal{F}, \|\cdot\|_{\Omega_R}, \frac{1}{N}) + \log(K \log(N)/\delta) \right)}{N} + r_j \sqrt{\frac{M_R \log \mathcal{N}(\mathcal{F}, \|\cdot\|_{\Omega_R}, \frac{1}{N})}{N}} \tag{B.84}
$$

By Bousquet (2002, Lemma 6.1), with probability at least $1 - \delta/(4K)$, for any $j \leq j_0, (\mathcal{S}, f) \in \mathcal{M}_k(r_j)$,

$$
\left| \mathbb{E}_{y \sim \mathbb{P}_k^Y, x \sim \widetilde{\mathbb{P}}_{\widehat{h}_k}(\cdot|y)}[\bar{\ell}(x, y, f, \mathcal{S})] - \frac{1}{N} \sum_{i=1}^{N} [\bar{\ell}(\widetilde{x}_i^k, \widetilde{y}_i^k, f, \mathcal{S})] \right| \lesssim \mathcal{R}_N(\Psi_k(r_j)) + r_j \sqrt{\frac{M_R \log(\frac{K}{\delta})}{N}} + \frac{M_R \log(\frac{K}{\delta})}{N}. \tag{B.85}
$$

Define $\widehat{r}_k^2 := \min\{r_j : r_j^2 \geq \mathbb{E}_{y \sim \mathbb{P}_k^Y, x \sim \widetilde{\mathbb{P}}_{\widehat{h}_k}(\cdot|y)} \mathbb{E}_{t, x_t} \|\widehat{f}_{\mathcal{S}}(x_t, y, t) - \widetilde{f}_{\mathcal{S}}(x_t, y, t)\|^2, j \leq j_0\}$. Then we have

$$
\left| \mathbb{E}_{y \sim \mathbb{P}_k^Y, x \sim \widetilde{\mathbb{P}}_{\widehat{h}_k}(\cdot|y)} \bar{\ell}(x, y, \widehat{f}_{\mathcal{S}}, \mathcal{S}) - \frac{1}{N} \sum_{i=1}^{N} \bar{\ell}(\widetilde{x}_i^k, \widetilde{y}_i^k, \widehat{f}_{\mathcal{S}}, \mathcal{S}) \right| \lesssim \mathcal{R}_N(\Psi_k(\widehat{r}_k)) + \widehat{r}_k \sqrt{\frac{M_R \log(\frac{K}{\delta})}{N}} + \frac{M_R \log(\frac{K}{\delta})}{N}. \tag{B.86}
$$

Since $\|\widetilde{x}_i^k\|_\infty \leq R$, by definition we also have $\widehat{f}_{\mathcal{S}} = \arg\min_{f \in \mathcal{F}} \sum_{k=1}^{K} \frac{\lambda_k}{N} \sum_{i=1}^{N} \bar{\ell}(\widetilde{x}_i^k, \widetilde{y}_i^k, f, \mathcal{S})$.

$$
\sum_{k=1}^{K} \lambda_k \mathbb{E}_{y \sim \mathbb{P}_k^Y, x \sim \widetilde{\mathbb{P}}_{\widehat{h}_k}(\cdot|y)} \bar{\ell}(x, y, \widehat{f}_{\mathcal{S}}, \mathcal{S}) \lesssim \sum_{k=1}^{K} \lambda_k \left[ \mathcal{R}_N(\Psi_k(\widehat{r}_k)) + \widehat{r}_k \sqrt{\frac{M_R \log(\frac{K}{\delta})}{N}} \right] + \frac{M_R \log(\frac{K}{\delta})}{N}. \tag{B.87}
$$

Therefore,

$$
\sum_{k=1}^{K} \lambda_k \mathbb{E}_{y \sim \mathbb{P}_k^Y, x \sim \widetilde{\mathbb{P}}_{\widehat{h}_k}(\cdot|y)} [\ell(x, y, \widehat{f}_{\mathcal{S}}) - \ell(x, y, \widetilde{f}_{\mathcal{S}})] \lesssim \sum_{k=1}^{K} \lambda_k \left[ \mathcal{R}_N(\Psi_k(\widehat{r}_k)) + \widehat{r}_k \sqrt{\frac{M_R \log(\frac{K}{\delta})}{N}} \right] + \frac{M_R \log(\frac{K}{\delta})}{N}. \tag{B.88}
$$

Combining the inequality above with Eqs. (B.72) and (B.84),

$$
\sum_k \lambda_k \widehat{r}_k^2 \lesssim \sum_{k=1}^{K} \lambda_k \left[ \mathcal{R}_N(\Psi_k(\widehat{r}_k)) + \widehat{r}_k \sqrt{\frac{M_R \log(\frac{K}{\delta})}{N}} \right] + \frac{M_R \log(\frac{K}{\delta})}{N}
$$
$$
\lesssim \sum_{k=1}^{K} \lambda_k \widehat{r}_k \sqrt{\frac{M_R(\log \mathcal{N}(\mathcal{F}, \|\cdot\|_{\Omega_R}, \frac{1}{N}) + \log(\frac{K}{\delta}))}{N}} + \frac{M_R \left( \log \mathcal{N}(\mathcal{F}, \|\cdot\|_{\Omega_R}, \frac{1}{N}) + \log(K \log(N)/\delta) \right)}{N}. \tag{B.89}
$$

Hence by Jensen's inequality,

$$\sum_{k=1}^{K} \lambda_k \widehat{r}_k \lesssim \sqrt{\frac{M_R(\log \mathcal{N}(\mathcal{F}, \|\cdot\|_{\Omega_R}, \frac{1}{N}) + \log(\frac{K}{\delta}))}{N}}. \tag{B.90}$$

We finally conclude that with probability no less than $1 - \delta/2$,

$$\sum_{k=1}^{K} \lambda_k \mathbb{E}_{y \sim \mathbb{P}_k^Y, x \sim \widetilde{\mathbb{P}}_{\widehat{h}_k}(\cdot|y)}[\ell(x, y, \widehat{f}_{\mathcal{S}}) - \ell(x, y, \widetilde{f}_{\mathcal{S}})] \lesssim \frac{M_R(\log \mathcal{N}(\mathcal{F}, \|\cdot\|_{\Omega_R}, \frac{1}{N}) + \log(\frac{K}{\delta}))}{N}. \tag{B.91}$$

Let $R = C \log^{\frac{1}{2}}(NKd_x M_1 M_0/\delta)$ for some large constant $C$, $T = \mathcal{O}(\log n), T_0 = \mathcal{O}\left(1/(n^2 \log^{d_x+1}(n))\right)$. Then $M_R \lesssim M_0^2 + d_x M_1^2 \log(\frac{NK}{\delta})$ and with probability at least $1 - \delta$, for any $\mathcal{S} \in \mathcal{S}_{\text{lin}}$,

$$\sum_{k=1}^{K} \lambda_k \mathbb{E}_{y \sim \mathbb{P}_k^Y, x \sim \widetilde{\mathbb{P}}_{\widehat{h}_k}(\cdot|y)}[\ell(x, y, \widehat{f}_{\mathcal{S}}) - \ell(x, y, \widetilde{f}_{\mathcal{S}})] \lesssim \left(M_0^2 + d_x M_1^2 \log(\frac{NK}{\delta})\right) \frac{\log \mathcal{N}(\mathcal{F}, \|\cdot\|_{\Omega_R}, \frac{1}{N}) + \log(\frac{K}{\delta})}{N}, \tag{B.92}$$

indicating that (recalling the definition of $\bar{\varepsilon}_{\text{apx}}$)

$$\mathbb{E}_{y \sim \mathbb{P}_k^Y, x \sim \widetilde{\mathbb{P}}_{\widehat{h}_k}(\cdot|y)} \mathbb{E}_{t, x_t}[\|\widehat{f}_{\mathcal{S}}(x_t, y, t) - \widehat{h}_k(x_t, y, t)\|^2] \leq \bar{\varepsilon}_{\text{apx}}^2(\mathcal{S})$$
$$+ \left(M_0^2 + d_x M_1^2 \log(\frac{NK}{\delta})\right) \frac{\log \mathcal{N}(\mathcal{F}, \|\cdot\|_{\Omega_R}, \frac{1}{N}) + \log(\frac{K}{\delta})}{N}. \tag{B.93}$$

According to Lemma B.3,

$$\sum_{k=1}^{K} \lambda_k \mathbb{E}_{y \sim \mathbb{P}_k^Y} \text{TV}(\mathbb{P}_{\widehat{f}_{\mathcal{S}}}(\cdot|y), \widetilde{\mathbb{P}}_{\widehat{h}_k}(\cdot|y)) \lesssim \bar{\varepsilon}_{\text{apx}}(\mathcal{S}) + \left(M_0 + M_1 d_x^{\frac{1}{2}} \log^{\frac{1}{2}}(\frac{NK}{\delta})\right) \sqrt{\frac{\log \mathcal{N}(\mathcal{F}, \|\cdot\|_{\Omega_R}, \frac{1}{N}) + \log(\frac{K}{\delta})}{N}}. \tag{B.94}$$

Finally we invoke Lemma B.5 and Lemma B.4 and conclude that

$$\sum_{k=1}^{K} \lambda_k \mathbb{E}_{y \sim \mathbb{P}_k^Y} \text{TV}(\mathbb{P}_{\widehat{f}_{\mathcal{S}}}(\cdot|y), \mathbb{P}_k(\cdot|y)) \lesssim \bar{\varepsilon}_{\text{apx}}(\mathcal{S})$$
$$+ \left(M_0 + M_1 d_x^{\frac{1}{2}} \log^{\frac{1}{2}}(\frac{NK}{\delta})\right) \left(\sqrt{\frac{\log \mathcal{N}(\mathcal{F}, \|\cdot\|_{\Omega_R}, \frac{1}{N})}{N}} + \sum_{k=1}^{K} \lambda_k \sqrt{\frac{\log \mathcal{N}(\mathcal{H}_k, \|\cdot\|_{\Omega_R}, \frac{1}{n^2}) + \log(\frac{K}{\delta})}{n}}\right). \tag{B.95}$$

$\square$

## B.4. Proof in Sec. 3.3

**Theorem B.9.** *Under Asps. 3.1 to 3.3 and the MDP setting, assume that $\mathcal{F}$ is convex. The following with probability no less than $1 - \delta$: for any $\mathcal{S} \in \mathcal{S}_{lin}$,*

$$\mathcal{S}\left(\left(V_{M_k}(\pi_k^*) - V_{M_k}(\pi_{\widehat{f}_{\mathcal{S}}})\right)_k\right) \lesssim \bar{\varepsilon}_{apx}(\mathcal{S})$$
$$+ \frac{M_0 + M_1 d_x^{\frac{1}{2}} \log^{\frac{1}{2}}(\frac{NK}{\delta})}{(1 - \gamma)^2} \left(\sqrt{\frac{\log \mathcal{N}(\mathcal{F}, \|\cdot\|_{\Omega_R}, \frac{1}{N})}{N}} + \sum_{k=1}^{K} \lambda_k \sqrt{\frac{\log \mathcal{N}(\mathcal{H}_k, \|\cdot\|_{\Omega_R}, \frac{1}{n^2}) + \log(\frac{K}{\delta})}{n}}\right) \tag{B.96}$$

*Proof.* Define the value function of $M$ under policy $\pi$ and initial state $s_0$ as

$$V_M(\pi, s_0) := \mathbb{E}\left[\sum_{t=0}^{\infty} \gamma^t r_M(s_t, a_t)\right], a_t \sim \pi(\cdot|s_t), s_{t+1} \sim \mathcal{T}_M(\cdot|s_t, a_t), \tag{B.97}$$
$$V_M(\pi) := \mathbb{E}_{s_0 \sim \rho_M}[V_M(\pi, s_0)].$$

Let $A_M^\pi(s,a) = Q_M^\pi(s,a) - V_M(\pi,s)$ be the advantage function of policy $\pi$ under environment $M$. Note that the reward function $r_M \in [-1,1]$, we have $|A_M^\pi(s,a)| \le \frac{2}{1-\gamma}$ for any $M, \pi$. According to performance difference lemma,

$$
\begin{aligned}
V_{M_k}(\pi_k^*) - V_{M_k}(\pi_{\widehat{f}_{\mathcal{S}}}) &= \frac{1}{1-\gamma} \mathbb{E}_{(s,a) \sim \mathbb{P}_k}[A_{M_k}^{\pi_{\widehat{f}_{\mathcal{S}}}}(s,a)] \\
&= \frac{1}{1-\gamma} \mathbb{E}_{s \sim \mathbb{P}_k} \left[ \mathbb{E}_{a \sim \pi_k^*(\cdot|s)}[A_{M_k}^{\pi_{\widehat{f}_{\mathcal{S}}}}(s,a)] - \mathbb{E}_{a \sim \pi_{\widehat{f}_{\mathcal{S}}}(\cdot|s)}[A_{M_k}^{\pi_{\widehat{f}_{\mathcal{S}}}}(s,a)] \right] \\
&\le \frac{2}{(1-\gamma)^2} \mathbb{E}_{s \sim \mathbb{P}_k} \mathrm{TV}(\pi_k^*(\cdot|s), \pi_{\widehat{f}_{\mathcal{S}}}(\cdot|s)) \\
&\le \frac{2}{(1-\gamma)^2} \left( \mathbb{E}_{s \sim \mathbb{P}_{M_k}^{\pi_{\widehat{h}_k}}} \mathrm{TV}(\pi_k^*(\cdot|s), \pi_{\widehat{f}_{\mathcal{S}}}(\cdot|s)) + \mathrm{TV}(\mathbb{P}_k, \mathbb{P}_{M_k}^{\pi_{\widehat{h}_k}}) \right)
\end{aligned}
\tag{B.98}
$$

By definition of total variation distance,

$$
\begin{aligned}
\mathrm{TV}(\mathbb{P}_k, \mathbb{P}_{M_k}^{\pi_{\widehat{h}_k}}) &= \sup_{\Omega \subseteq S} |\mathbb{P}_k(\Omega) - \mathbb{P}_{M_k}^{\pi_{\widehat{h}_k}}(\Omega)| \\
&= \sup_{\Omega \in S} \left| \mathbb{E}_{(s,a) \sim \mathbb{P}_k} \mathbb{1}_{s \in \Omega} - \mathbb{E}_{(s,a) \sim \mathbb{P}_{M_k}^{\widehat{h}_k}} \mathbb{1}_{s \in \Omega} \right| \\
&\le \mathbb{E}_{s \sim \mathbb{P}_k} \mathrm{TV}(\pi_k^*(\cdot|s), \pi_{\widehat{h}_k}(\cdot|s)) \\
&\lesssim \sqrt{L_{\mathbb{P}_k}(\widehat{h}_k)} + \frac{1}{N}.
\end{aligned}
\tag{B.99}
$$

Here in the last inequality we invoke Lemma B.3 and plug in the definition of $T, T_0$. Finally we apply Thm. B.8 and Lemma B.5 (note that here the ground truth distribution is $\mathbb{P}_{\widehat{h}_k}$) to get

$$
\begin{aligned}
\sum_{k=1}^K \lambda_k (V_{M_k}(\pi_k^*) - V_{M_k}(\pi_{\widehat{f}_{\mathcal{S}}})) &\lesssim (1-\gamma)^{-2} \bar{\varepsilon}_{\mathrm{apx}}(\mathcal{S}) \\
&+ \frac{M_0 + M_1 d_x^{\frac{1}{2}} \log^{\frac{1}{2}}(\frac{NK}{\delta})}{(1-\gamma)^2} \left( \sqrt{\frac{\log \mathcal{N}(\mathcal{F}, \|\cdot\|_{\Omega_R}, \frac{1}{N})}{N}} + \sum_{k=1}^K \lambda_k \sqrt{\frac{\log \mathcal{N}(\mathcal{H}_k, \|\cdot\|_{\Omega_R}, \frac{1}{n^2}) + \log(\frac{K}{\delta})}{n}} \right).
\end{aligned}
\tag{B.100}
$$

$\square$

