# OpenReview forum: "Multi-Objective Learning for Diffusion Models: A Statistical Theory under Semi-Supervised Learning"
_ICML.cc/2026/Conference — ICML 2026 regular_

### Official Review · Reviewer_4tQf · 2026-03-01

**Soundness:** 3
**Presentation:** 3
**Significance:** 3
**Originality:** 3
**Overall Recommendation:** 4
**Confidence:** 4

**Summary:**

The paper studies a two-stage approach for multi-objective learning in a semi-supervised setting. The main contribution of the paper is to develop non-asymptotic error bounds for the proposed objective.

**Compliance With Llm Reviewing Policy:**

Affirmed.

**Final Justification:**

I read the responses and I am satisfied.

**Key Questions For Authors:**

a. Could the authors explain in more detail how the first inequality in (A.52) is derived?

b. In Theorem A.5, inequality (A.54), could the authors explicitly justify that after conditioning on the truncation event and on $\hat{h}_k$, the second-stage data are still i.i.d. such that this upper bound holds for the empirical process?

**Limitations:**

yes

**Strengths And Weaknesses:**

Strength:

    1. The semi-supervised multi-objective learning setting is interesting and relevant to real-world applications.
    2. This appears to be the first theoretical work for diffusion models in a semi-supervised setting with multiple objectives.

Weakness:

1. Major concerns:

    a.  The zero score approximation error condition (Assumption 3.3) in the first stage is stronger than assumptions made in related theoretical works (e.g., [1][2][3][4]). The function $\mathcal{H}_k$ should be chosen to balance the statistical error and the approximation error between the class $\mathcal{H}_k$ and the ground truth by assuming that the ground truth belongs to a Holder class or a Sobolev space. Combined with assumption 3.2, it would be natural to consider the function class $\mathcal{H}_k$ as a neural network class with a bounded Lipschitz constant.

    b.  Unlike some previous statistical works in semi-supervised settings (e.g., [5][6]), the error bound in the paper seems not useful when the number of unlabeled samples is small (or fixed), especially when there is no unlabeled data. Furthermore, in my opinion, it would be helpful to discuss why the labeled data are not used in the second stage; for example, whether this is needed to construct independence between $\hat{h}_k$ and unlabeled data. In my view, cross-fitting (e.g., [6][7]) might address this and may lead to an improved upper bound.

    c.  Line 918, the proof uses a tail probability bound for the truncation of $\tilde{x}_i^k$

under $P_{\hat{h}_k}$.

However, assumption 3.1 in the main paper is imposed on $P_{k}$, and I do not find the derivation of the tail probability in Line 918. Could you please provide a derivation for this step?






Reference

[1] Huang, J., Jiao, Y., Li, Z., Liu, S., Wang, Y., and Yang, Y. (2022). An error analysis of generative adversarial
networks for learning distributions. Journal of Machine Learning Research, 23(116):1–43.

[2] Zhou, Z. and Liu, W. (2025). An error analysis of flow matching for deep generative modeling. In Forty-second
International Conference on Machine Learning.

[3] Chen, M., Huang, K., Zhao, T., and Wang, M. (2023). Score approximation, estimation and distribution
recovery of diffusion models on low-dimensional data. In Krause, A., Brunskill, E., Cho, K., Engelhardt,
B., Sabato, S., and Scarlett, J., editors, Proceedings of the 40th International Conference on Machine
Learning, volume 202 of Proceedings of Machine Learning Research, pages 4672–4712. PMLR.

[4] Gao, Y., Huang, J., Jiao, Y., and Zheng, S. (2024). Convergence of continuous normalizing flows for learning
probability distributions. arXiv preprint arXiv:2404.00551.

[5] Zhang, A., Brown, L. D., and Cai, T. T. (2019). Semi-supervised inference: General theory and estimation of means, The Annals of Statistics, 47(5), 2538-2566.

[6] Wen, M., Jia, Y., Ren, H., Wang, Z., and Zou, C. (2025). Semi-supervised distribution learning. Biometrika,
112(1):asae056.

[7] Chernozhukov, V., Chetverikov, D., Demirer, M., Duflo, E., Hansen, C., Newey, W., and Robins, J. (2018).
Double/debiased machine learning for treatment and structural parameters. The Econometrics Journal,
21(1)

---

> ### Author Rebuttal · Authors · 2026-03-31
>
> **W1.** Assumption 3.3 is too strong. It would be natural to consider the function class $\mathcal{H}_k$ as a neural network class with a bounded Lipschitz constant.
>
> **A1.** This work primarily focuses on statistical rates rather than (nonparametric) approximation guarantees, and Assumption 3.3 follows prior work on multi-objective learning (Wegel et al., 2025). The central message of the paper is that pseudo-labels generated by specialists can be used to distill a generalist model. Accordingly, extending our analysis to more realistic function classes—e.g., taking $\mathcal{H}_k$ to be a neural network class with bounded Lipschitz constant—and establishing end-to-end guarantees remains an interesting direction for future work.
>
> **W2.** The error bound in the paper seems not useful when the number of unlabeled samples is small. It would be helpful to discuss why the labeled data are not used in the second stage.
>
> **A2.** Thanks for the reviewer's suggestion. Our work focuses on the regime where unlabeled data are abundant and inexpensive to collect, and we therefore naturally assume access to a large amount of unlabeled samples in Stage 2. In practice, one could certainly incorporate labeled data in Stage 2 as well. From a theoretical perspective, however, if labeled data were also reused in Stage 2, the dependence between these data and the learned specialists $\widehat{h}_k$ would introduce additional technical difficulties into the analysis. In particular, cross-fitting may be a good idea to address this issue. We will discuss this in the revised version and leave this as an interesting direction for future work.
>
> **W3.** Line 918, the proof uses a tail probability bound for the truncation of $\tilde{x}\_{i}^{k}$ under $P_{\hat{h}_k}$. However, assumption 3.1 in the main paper is imposed on $P\_k$.
>
> **A3.** Thanks for pointing out this techinical issue. We agree that it is not valid to directly apply sub-gaussian tail for $P\_{\widehat{h}\_k}(\cdot|y)$ without further assumptions. Notice that The truncation argument is introduced for technical convenience, ensuring that $\tilde{x}\_{i}^{k}$ lies in a bounded domain $\Omega=B_\infty(R)$ and thus the individual score-matching loss objective can be bounded by $poly(R)$.
>
> There are two approaches to solve this issue:
>
> (1) a naive approach is to assume that $P_{h_k}$ is sub-gaussian for all $h_k\in\mathcal{H}_k$ (or at least in a neighborhood of $s_k^*$);
>
> (2) a more technical approach is to manually truncate $\tilde{x}\_{i}^{k}$ via rejection sampling and thus $\tilde{x}\_{i}^{k}\sim \tilde{P}\_{\widehat{h}\_k}(\cdot|\tilde{y}\_{i}^{k})\propto P\_{\widehat{h}\_k}(\cdot|\tilde{y}\_{i}^{k})\cdot 1\_{|\cdot|\_\infty\leq R}$, where $R$ is a user defined large constant. Note that $P_k(\cdot|y)$ only has $O(\exp(-CR^2))$ measure outside $\Omega=B_\infty(R)$ and $TV(P\_k,P\_{\widehat{h}\_k})$ is small. Also notice that for two density functions $p,q$ with $TV(p,q)\leq \epsilon$ and $q\_\Omega=\int\_\Omega q(x) dx\geq 1-\delta$, we have $p_\Omega\geq q_\Omega -\epsilon\geq 1-\delta-\epsilon$, and thus
> $\int |\frac{p\cdot 1_\Omega}{p_\Omega}-q|\leq \int |\frac{(p-q)\cdot 1_\Omega}{p_\Omega}|+\int q(\frac{1}{p_\Omega}-1)\leq \frac{\epsilon}{1-(\epsilon+\delta)}+\frac{1}{1-(\epsilon+\delta)}-1\leq 2\delta+4\epsilon$ when $\delta+\epsilon\leq \frac{1}{2}$. Therefore, $TV(\tilde{P}_{\widehat{h}_k},P_k)$ is still small. In this case, the proofs in Appendix A.5 is still valid. We will bridge this gap in the revised version.
>
> **W4.** How is (A.52) is derived?
>
> **A4.** We note that $\Big|E\_{(x,y)\sim P\_k}[\ell(x,y,f)-\ell(x,y,\widehat{h}\_k)]-E\_{y\sim P\_k^Y,x\sim P\_{\widehat{h}\_k}(\cdot|y)}[\ell(x,y,f)-\ell(x,y,\widehat{h}\_k)]\Big|$ can be decomposed into three terms: $\Big|E\_{(x,y)\sim P\_k}[\ell(x,y,f)-\ell(x,y,\widehat{h}\_k)]\cdot 1\_{|x|\_{\infty}\leq R}-E\_{y\sim P\_k^Y,x\sim P\_{\widehat{h}\_k}(\cdot|y)}[\ell(x,y,f)-\ell(x,y,\widehat{h}\_k)]\cdot 1\_{|x|\_{\infty}\leq R}\Big|$, $\Big|E\_{(x,y)\sim P\_k}[\ell(x,y,f)-\ell(x,y,\widehat{h}\_k)]\cdot 1_{|x|\_{\infty}> R}$, and $E\_{y\sim P\_k^Y,x\sim P\_{\widehat{h}\_k}(\cdot|y)}[\ell(x,y,f)-\ell(x,y,\widehat{h}\_k)]\cdot 1\_{|x|\_{\infty}\leq R}\Big|$.
>
> Eq (A.51) shows that when $|x|\_\infty\leq R$, $|\ell(x,y,f)-\ell(x,y,\widehat{h}\_k)|\leq M\_R$, and therefore the first term can be bounded by $M\_R E\_{y\sim P\_k^Y}\mathrm{TV}(P\_k(\cdot|y),P\_{\widehat{h}\_k}(\cdot|y))$ by definition of TV distance. We will elaborate this in the revised version.
>
> **W5.** In (A.54), justify that after conditioning on the truncation event and on $\hat{h}_k$, the second-stage data are still i.i.d..
>
> **A5.** Since $\tilde{y}\_i^k$ are i.i.d. sampled and independent of $\widehat{h}\_k$, and $\tilde{x}^k\_i\sim P\_{\widehat{h}\_k}(\cdot|\tilde{y}\_i^k)$ are independent from each other conditioned on $\widehat{h}\_k$, there the second stage data is still i.i.d. and eq (A.54) holds. We will elaborate this in the revised version.

---

> > ### Author Rebuttal · Reviewer_4tQf · 2026-04-01
> >
> > My questions have been addressed.

---

### Official Review · Reviewer_paB9 · 2026-03-09

**Soundness:** 4
**Presentation:** 3
**Significance:** 3
**Originality:** 3
**Overall Recommendation:** 5
**Confidence:** 4

**Summary:**

The paper studies conditional generative modeling with diffusion models in a semi-supervised learning setting, where more (unlabeled) conditions are available than paired samples. This transfers ideas for prediction from Wegel et al. (2025) to this setting. The paper introduces a two-stage procedure that fits expert models to the individual distributions and then uses them to create a pseudo-labeled data set to train a generalist model. It provides non-asymptotic guarantees on the distribution approximation error from finite samples, and extends the analysis to non-iid sampling.

**Compliance With Llm Reviewing Policy:**

Affirmed.

**Final Justification:**

Overall, I view this paper's main contributions as theoretical and interesting. I do share the sentiment of the other reviewers that the experimental section needs a revision, but the additional experiments provided during the rebuttal make me think that this does not require a resubmission and that this work can be published.

**Key Questions For Authors:**

1. What is the intuition that unlabeled data helps here?
2. Did you test your algorithm also on other conditional data generation, e.g., in text-to-image generation? In particular, as you mention this as one of the examples on page 4.
3. Does your theory require $\mathcal{H}_k$ to be subsets of $\mathcal{F}$? If so, in  Assumption 3.2 its somewhat redundant. If not, should stage 2 of the algorithm not induce an approximation error?
4. Could the authors clarify Assumption 3.3: In Wegel et al. (2025), realizability is not assumed; indeed, labels are allowed to be inherently noisy. Instead, it is effectively assumed that the conditional expectation of $Y|X$ is contained in $\mathcal{H}_k$. It is the inherent noise that also requires the loss to be a Bregman loss. Here it seems that you actually require realizability (and so there is no noise). Is that necessary? Or is it because of the generative setting?  Can the analysis deal with the non-realizable setting/how does it relate to approximation errors in Cor 3.4, Thm 3.5?
5. In this work the objectives are square losses on the score matching objective and then transforms that into bounds on the TV distance. In Wegel et al. (2025), the framework somewhat naturally uses divergences; could one also aim to use that directly?
6. In Remark 1; is the $1/\varepsilon^4$ vs $1/\varepsilon^2$ really due to the „specific structure of the matching objective“, and not simply due to the fact that in your proof, you use a localized bound for the first step (which in Wegel et al. (2025) is done in Thm 2)?
7. In Thm 3.5, where exactly does the speed-up come from? Is it also applying localization to the second stage of the algorithm? One or two sentences of intuition would help.
8. In the experiments, what exactly is the baseline? Just training directly on the labeled data?

**Limitations:**

yes

**Strengths And Weaknesses:**

*Strengths:*
The work nicely merges ideas from Wegel et al. (2025) with the analysis techniques developed in Cheng et al. (2025). The setting and analysis make sense, and the authors make non-trivial contributions in terms of formulating the problem and proving error bounds. In doing so, the authors are quite sound, and present the results with the necessary rigor, while omitting unnecessary detail.

*Weaknesses:*
The experiments are on the shorter side. Moreover, I think (the intuition behind) the stated examples, in particular text-to-image generation and robotic control, are not discussed very carefully. What are the concrete use cases of the multi-objective viewpoint here?

*Small remarks:*
- In Equation (2.1), $\overline{W}_t$ is used without being introduced.
- It would be good to spend more time in Section 2.1 to introduce some of these objects with more detail. In particular, I am for instance somewhat confused as to what stopping time is used in the end to achieve these results.
- Similar to the updated conference version of Wegel et al. (2025), perhaps you require monotonicity of the scalarization?
- I think Figure 1 could substantially be improved to convey how the algorithm works (the big yellow robot arm is not telling me much)
- Do the constants in Assumption 3.1 depend on $y$? Its a bit ambiguous.

---

> ### Author Rebuttal · Authors · 2026-03-31
>
> ### Writing issues
>
> Thanks for the suggestions and we will address these issues accordingly in the revised version. Specidically, the early stopping time is $T\_0=O(\frac{1}{N^2 polylog(N)})$. And the constant doesn't depend on $y$.
>
> ### Questions
> **W1.** What is the intuition that unlabeled data helps here?
>
> **A1.** Our algorithm proceeds in two stage. In the first stage, we use the limited paired data to train a specialist model for each task $k \in [K]$. In the second stage, we use these trained specialists to generate pseudo-labels for the unlabeled data. We then train a generalist model on the combination of the pseudo-labeled data and the paired data from the first stage.
> - As shown in our theoretical results, this two-stage method succesfully reduces the sample complexity of multi-objective learning from dependance on the generalist model class $\mathcal F$ to $\{\mathcal H_k \}$, whose complexity is substantially smaller than that of $\mathcal F$.
>
>
>
> **W2.** Did you test your algorithm also on other conditional data generation, e.g., in text-to-image generation? In particular, as you mention this as one of the examples on page 4.
>
> **A2.** We conducted additional experiments on the robotics and image restoration tasks. Please refer to the response to W2 of reviewer V9TH.
>
> **W3.** Does your theory require $\mathcal H_k$ to be subsets of $\mathcal F$ ? If so, in Assumption 3.2 its somewhat redundant. If not, should stage 2 of the algorithm not induce an approximation error?
>
> **A3.** Thanks for the reviewer's suggestion. We do require $\mathcal H_k$ to be subsets of $\mathcal F$, and we will revise the statement of Assumption 3.2 accordingly. But we would like to mention that we don't require realizability of $\mathcal{F}$ on all objectives.
>
> **W4.** Could the authors clarify Assumption 3.3: is it because of the generative setting? Can the analysis deal with the non-realizable setting？ How does it relate to approximation errors in Cor 3.4, Thm 3.5?
>
> **A4.** In Wegel et al. (2025, Eq (3)), the conditional expectation of $Y | X$ has to be contained in $\mathcal H_k$. However, the Bregman divergence in Wegel et al. (2025) cannot characterize discrepancy between distributions. Since we focus on generative model setting instead of mere prediction error, we have to impose realizability at the distribution level, i.e., Assumption 3.3. More generally, one could allow an approximation error $\epsilon$ for the specialist class, in which case the final statistical rates would additionally depend on $\epsilon$. However, this extension is orthogonal to the main message of the paper, namely, that pseudo-labels generated by specialists can be used to distill a generalist model. We highlight that this approximation error is for single objective, different from that in Cor 3.4, Thm 3.5, which is the error for multi-objective.
>
> **W5.** In this work the objectives are square losses on the score matching objective and then transforms that into bounds on the TV distance. In Wegel et al. (2025), the framework somewhat naturally uses divergences; could one also aim to use that directly?
>
> **A5.** Our work focuses on score-based diffusion models and therefore naturally centers on the score-matching objective. By contrast, Wegel et al. (2025) studies prediction problems only, where the Bregman divergence serves to characterize prediction error (e.g., mean squared error), rather than the actual discrepancy between probability distributions. even with This is fundamentally different from the generative modeling setting considered in our work.
>
> **W6.** In Remark 1; is the $1 / \epsilon^4$ vs $1 / \epsilon^2$ not simply due to the localized bound for the first step (see Wegel et al. (2025) Thm 2)?
>
> **A6.** In Thm 3.2, we indeed employ a localization argument to obtain a sharper rate. However, the localized result in Wegel et al. (2025, Thm 2) relies on additional assumptions, including star-shapedness of the function class and strong convexity of the loss. By contrast, our analysis in Theorem 3.2 does not require these assumptions, primarily dues to the score-matching structure.
>
> **W7.** In Thm 3.5, where exactly does the speed-up come from? Is it also applying localization to the second stage of the algorithm?
>
> **A8.** When the scalarization funciton is linear, the resulting scalarized score-matching loss remains a time-averaged quadratic objective, and hence the minimizer stability property continues to hold; see Eq.(A.71). Combined with the localized bounds in Eqs. (A.83) and (A.87), this yields the reverse inequality in Eq. (A.88), which in turn leads to a sharper control of the localization radius and therefore an improved sample complexity rate.
>
> **W9.** In the experiments, what exactly is the baseline? Just training directly on the labeled data?
>
> **A9.** Yes, for the multi-task baseline, we trained the generalist with the paired data without the pseudo samples.

---

> > ### Author Rebuttal · Reviewer_paB9 · 2026-04-02
> >
> > Thank you for the detailed response. Also based on the other reviews and responses, and in particular because of the additional experiements presented to reviewer V9TH whic address my main reservation, i retain my positive view of this work as well as my score.

---

### Official Review · Reviewer_a3Rr · 2026-03-13

**Soundness:** 2
**Presentation:** 2
**Significance:** 2
**Originality:** 2
**Overall Recommendation:** 3
**Confidence:** 5

**Summary:**

To address the challenges of scarce paired samples and high statistical costs for training generalist models in multi-objective learning (MOL) with diffusion models, this work proposes a semi-supervised two-stage training pipeline: first, lightweight specialist models are trained on limited paired data, and then a generalist model is distilled from pseudo-samples generated by these specialists. The authors derive non-asymptotic generalization bounds, proving that the paired-sample complexity depends only on the capacity of the specialist model classes rather than the generalist class. The theory is further extended to diffusion policies for sequential decision-making to tackle distribution shift issues, and the theoretical validity is verified via experiments on robot manipulation tasks.

**Compliance With Llm Reviewing Policy:**

Affirmed.

**Final Justification:**

Lemma 3.1 and Theorem 3.2 present the foundation for the approximation loss bound. However, the connection between them is somewhat loose. I suggest that the authors quantify their results to better understand which variables and factors influence the tightness of the bound. The current presentation follows a typical approach and does not offer substantial new insights to the learning community. Additionally, the experimental evaluation appears insufficient.

**Key Questions For Authors:**

1、The paper claims to achieve improved paired-sample complexity compared to Zhang et al. and Wegel et al. Could the authors provide a quantitative comparison table to clearly demonstrate the practical advantages of the proposed theory?

2、The experiments in this paper are limited to a single robotic manipulation environment.Have the authors considered evaluating the method on other diffusion model applications, such as image generation and image restoration, to demonstrate the generalizability of the proposed framework?

3、The theory highlights the impact of the number of specialist models and pseudo-samples on performance. Could the authors conduct experiments to analyze how varying the number of specialists and pseudo-samples affects the performance of the generalist model, thereby validating the theoretical conclusions on sample complexity?

4、The necessity of the pseudo-sample distillation step is not empirically verified in the paper.What is the performance difference between the proposed two-stage training pipeline and the baseline method that trains the generalist model directly on the available paired samples without generating pseudo-samples?

**Limitations:**

yes

**Strengths And Weaknesses:**

### Strengths
1. The problem of scarce paired samples and abundant conditional data is a common scenario in industry, making this work strongly application-oriented.
2. This paper establishes a semi-supervised statistical theory for multi-objective learning with diffusion models, demonstrating notable innovation and theoretical value.
3. It proves that the paired-sample complexity of MOL for diffusion models depends solely on the specialist models.
4. The theoretical analysis outperforms existing related work: compared to the semi-supervised MOL study for prediction tasks (Wegel et al., 2025), this work achieves a sharper paired-sample complexity for the score-matching objective without imposing strong star-shaped structural assumptions on the model classes, thus enjoying wider applicability.


### Weaknesses

##### Theoretical Issues
Under the framework of semi-supervised multi-objective learning for diffusion models, it is proven that to achieve an error bound $\varepsilon_k \le \varepsilon$, the required **paired (labeled) sample complexity is $\tilde{O}\!\left(\frac{K C_{\mathcal{H}}}{\varepsilon^2}\right)$**, determined solely by the complexity $C_{\mathcal{H}}$ of the lightweight specialist model classes, independent of the complexity $C_{\mathcal{F}}$ of the large-capacity generalist class; the sample complexity for unlabeled conditioning variables is $\tilde{O}\!\left(\frac{K C_{\mathcal{F}}}{\varepsilon^2}\right)$. In contrast, the fully supervised baseline (Zhang et al., 2024) requires $\tilde{O}\!\left(\frac{K C_{\mathcal{F}}}{\varepsilon^2}\right)$ paired samples, while Wegel et al. (2025) requires $\tilde{O}\!\left(\frac{K C_{\mathcal{H}}}{\varepsilon^4}\right)$ paired samples for prediction tasks.

1. Overly strict assumptions: The theoretical analysis heavily relies on Assumption 3.3 (realizability of the specialist model classes), which limits practical applicability. If this assumption is violated, pseudo-samples will contain systematic biases that propagate to the generalist model, invalidating the theoretical conclusions.
2. The paper lacks a quantitative tabular comparison of sample complexity, model assumptions, and applicable scenarios with related works such as Wegel et al. (2025) and Zhang et al. (2024).

##### Experimental Issues
The method is only validated on a single robot manipulation scenario, and the absence of ablation experiments prevents a thorough verification of the necessity of each component:
1. The method is not validated on other classic diffusion model scenarios (e.g., text-to-image synthesis, image inpainting), restricting the proof of its generalizability.
2. The impact of the number of specialist models and pseudo-samples on generalist model performance is not evaluated, making it impossible to quantify the theoretical conclusions on sample complexity.
3. No ablation of core components of the two-stage training (e.g., removing pseudo-samples and training the generalist directly on paired samples) is performed, failing to fully demonstrate the necessity of pseudo-sample distillation.

---

> ### Author Rebuttal · Authors · 2026-03-31
>
> **W1.** Overly strict assumptions: The theoretical analysis heavily relies on Assumption 3.3 (realizability of the specialist model classes), which limits practical applicability. If this assumption is violated, pseudo-samples will contain systematic biases that propagate to the generalist model, invalidating the theoretical conclusions.
>
> **A1.** We note that realizability is a standard assumption in theoretical analyses of multi-objective learning [1,2]. More generally, one could allow an approximation error $\epsilon$ for the specialist class, in which case the final statistical rates would additionally depend on $\epsilon$. Indeed, if the specialist class incurs a large approximation error, then the resulting pseudo-labels will inevitably be biased and may degrade the distillation of the generalist. Nevertheless, each specialist only needs to model a single objective, which is typically substantially easier than solving all objectives simultaneously, especially in robotics applications. For this reason, we believe Assumption~3.3 is both reasonable and practically meaningful for the present analysis.
>
> **W2.** The paper lacks a quantitative tabular comparison of sample complexity, model assumptions, and applicable scenarios with related works such as Wegel et al. (2025) and Zhang et al. (2024).
>
> **A2.** We highlight that our analysis concerns **score-based generative model**, whereas previous works Wegel et al. (2025) and Zhang et al. (2024) only study prediction problems.These are fundamentally different settings: generative modeling is typically substantially more challenging than standard prediction, and the analysis of score matching objectives is also considerably more involved than that of simple convex losses in Wegel et al. (2025). As a result, our sample complexity is not directly comparable to those in these prior works. Moreover, our Assumption 3.1, 3.2 are weaker than Assumption 1 in Wegel et al. (2025), which imposes strong convexity, boundedness, etc. on the loss function. And our Assumption 3.3 is equivalent to the corresponding assumption in in Wegel et al. (2025)
>
> **W3.** The method is not validated on other classic diffusion model scenarios (e.g., text-to-image synthesis, image inpainting).
>
> **A3.** We conducted additional experiments on the robotics and image restoration tasks. Please refer to the response to W2 of reviewer V9TH.
>
> **W4.** The impact of the number of specialist models and pseudo-samples on generalist model performance is not evaluated, making it impossible to quantify the theoretical conclusions on sample complexity.
>
> **A4.** In this work, we treat the number of specialist models as fixed. Due to limited time, we were only able to conduct an ablation study on the effect of the number of pseudo-samples on the PickCube task. The evaluation is still ongoing, and we will update the results in the anonymous repository: https://anonymous.4open.science/r/MOL-Rebuttal-F9F7 once they are available.
>
> **W4.** No ablation of core components of the two-stage training (e.g., removing pseudo-samples and training the generalist directly on paired samples) is performed.
>
> **A4.** The multi-task baseline train the generalist directly on paired data without the pseudo samples.
>
> [1] Wegel, Tobias, et al. "On the sample complexity of semi-supervised multi-objective learning." arXiv preprint arXiv:2508.17152 (2025).
>
> [2] Tripuraneni, Nilesh, Michael Jordan, and Chi Jin. "On the theory of transfer learning: The importance of task diversity." Advances in neural information processing systems 33 (2020): 7852-7862.

---

> > ### Author Rebuttal · Reviewer_a3Rr · 2026-04-03
> >
> > Thank you for your reply addressing part of my comments. In light of the experimental results, however, I hold the view that this paper is not adequately ready for acceptance by ICML.

---

> > > ### Author Response · Authors · 2026-04-06
> > >
> > > We thank the reviewer again for the helpful feedback, which motivated us to further strengthen the empirical evaluation. In response, we have added experiments on an image inpainting task and provided further analysis on pseudo-sample scaling.
> > >
> > > > Additional results on image inpainting
> > >
> > > Specifically, we evaluate our method on CelebA-HQ [1] using three representative mask types from RePaint [2]: Wide, Face Mask, and Alternative Lines. The goal is to reconstruct the original image from the masked input. For each specialist model, we use 10k labeled images for training and generate pseudo-targets for the remaining 18k masked images using the trained specialist. The proposed semi-supervised generalist is then trained on the union of the labeled and pseudo-targeted data, resulting in 84k training images in total, while the multi-task baseline is trained on 30k labeled images. Both models are trained for 100k iterations and evaluated on 2,000 test images per mask type using Peak Signal-to-Noise Ratio (PSNR) and Structural Similarity Index Measure (SSIM), where higher values indicate better reconstruction quality.
> > >
> > >   The results are summarized below:
> > >
> > >   | Mask | Ours (Semi-Supervised Learning) | Multi-Task Baseline |
> > >   |---|---:|---:|
> > >   | Alt. Lines | **23.495 / 0.784** | 23.078 / 0.767 |
> > >   | Face Mask | **21.994 / 0.742** | 21.726 / 0.729 |
> > >   | Wide | **21.450 / 0.736** | 21.227 / 0.727 |
> > >
> > > As shown above, our method consistently outperforms the baseline across all metrics and mask types. These results demonstrate that our framework effectively leverages unlabeled data to improve reconstruction quality in inpainting, providing strong evidence of its generalizability beyond the tasks previously explored.
> > >
> > > > Impact of Pseudo-Sample Scaling (W4)
> > >
> > > We also refer the reviewers to our response for W4, where we conducted experiments on the impact of the number of pseudo-samples on on a robot manipulation task (PickCube). Specifically, we vary the number of pseudo-labeled trajectories, $N$ in {0, 400, 800, 1600} $N = 0$ corresponds to the multi-task baseline and $N = 1600$ corresponds to the semi-supervised result reported in the rebuttal.
> > >
> > > | Level         | N = 0 | N = 400 | N = 800 | N = 1600 |
> > > |---------------|------:|--------:|--------:|---------:|
> > > | Level 0       |   25% |      49% |       65% |        69% |
> > > | Level 1       |   26% |      45%  |       50% |      49% |
> > >
> > > These results show that adding pseudo-labeled data substantially improves performance over the multi-task baseline, especially at Level 0, where performance increases steadily as the number of pseudo-samples grows. For the Level 1 task, we observe a clear improvement from $N = 0$ to $N = 800$, after which the gains saturate. Overall, this ablation supports our claim that pseudo-labeled data plays an important role in improving task performance, while also suggesting that the marginal benefit may diminish beyond a certain scale.
> > >
> > > We hope these additional results help clarify the empirical strength of our method, and we respectfully ask the reviewer to reconsider the decision in light of this expanded evidence.
> > >
> > > **References**
> > >
> > > [1] Karras, Tero, et al. "Progressive growing of gans for improved quality, stability, and variation." arXiv preprint arXiv:1710.10196 (2017).
> > >
> > >
> > > [2] Lugmayr, Andreas, et al. "Repaint: Inpainting using denoising diffusion probabilistic models." Proceedings of the IEEE/CVF conference on computer vision and pattern recognition. 2022.

---

### Official Review · Reviewer_V9TH · 2026-03-18

**Soundness:** 4
**Presentation:** 3
**Significance:** 3
**Originality:** 3
**Overall Recommendation:** 3
**Confidence:** 3

**Summary:**

The article addresses the significant number of samples required when training multi-purpose conditional diffusion models. To reduce the need for costly labeled data, the authors suggest a semi-supervised learning approach with two stages. In the first stage, lightweight specialist models are trained on a small amount of paired data for each distinct objective. In the second stage, these specialists are used to generate pseudo-samples for a large set of unlabeled data, which are then used to distill the knowledge into a high-capacity generalist model. The core contribution is theoretical.

**Compliance With Llm Reviewing Policy:**

Affirmed.

**Final Justification:**

My concerns of experimental validation still stand. The authors have done a great job of exploring the theoretical direction of their work, but the experiments don't meet the bar for ICML in my view.

**Key Questions For Authors:**

Why did the authors test on only a Stacking robotics task? Other environments could have been considered.

**Limitations:**

The authors mention text-to-image models as one example where semi-supervised learning is helpful. However, experiments did not include text-to-image. They are limited to a specific robotics setting of stacking a cube. I find the experiments section of the work too limited.

**Strengths And Weaknesses:**

- The theoretical framework is rigorous and technically sound.
- The adaptation of MOL bounds to the specific score-matching objectives of diffusion models is non-trivial, well-executed, and original.
- The theoretical results rely heavily on Assumption 3.3 (Realizability), which asserts that the ground truth score function lies perfectly within the lightweight specialist class.
- The presentation is clear
- There are very few experiments

---

> ### Author Rebuttal · Authors · 2026-03-31
>
> **W1**. The theoretical results rely heavily on Assumption 3.3 (Realizability).
>
> **A1**. We note that realizability is a standard assumption in theoretical analyses of multi-objective learning [1,2]. More generally, one could allow an approximation error $\epsilon$ for the specialist class, in which case the final statistical rates would additionally depend on $\epsilon$. However, this extension is orthogonal to the main message of the paper, namely, that pseudo-labels generated by specialists can be used to distill a generalist model.
>
> **W2**. Additional experiments
>
> In addition to the StackCube task in the submission, we provide two additional results on robot manipulation and Text-to-Image generations respectively.
>
> - Following the same pipeline as StackCube, we verify our algorithm on PickCube, where the robot is asked to pick up a cube from the table. Starting from the baseline PickCube-L0, we instantiate three additional tasks using the randomization tools in Section 5: PickCube-L1-Table, PickCube-L1-Ground, and PickCube-L1-Wall. Thus, we consider $K = 4$ tasks in total. For each task, we train a specialist on $100$ expert demonstrations, and then use the specialists to generate $400$ additional pseudo-labeled trajectories per task. The final generalist is trained on the union of expert demonstrations and pseudo-labeled trajectories. The data available to each model is summarized below.
>
>     | Model | Supervised data | Pseudo-labeled data | Scope |
>     |---|---|---|---|
>     | Specialist for task $k$ | Task $k$ only ($n = 100$) | None | Per-task specialist |
>     | Multi-task baseline | All $K$ tasks ($Kn= 400$) | None | Joint training on labeled data only |
>     | Semi-supervised learning (ours) | All $K$ tasks ($Kn = 400$) | All $K$ tasks ($N = 1600$) | Joint training on labeled + pseudo-labeled data |
>
>     We report the success rate on four different randomization levels of PickCube. Levels 2 and 3 are considered as out-of-distribution (OOD) evaluations as they are not included in the training data. The result is consistent with the StackCube experiment that shows our method outperforms the multi-task baseline.
>
>
>     | Level         | Semi-supervised learning | Multi-task Baseline |
>     |---------------|--------------------------:|--------------------:|
>     | Level 0       | 69%                       | 25%                 |
>     | Level 1       | 49%                       | 26%                 |
>     | Level 2 (OOD) | 41%                       | 35%                 |
>     | Level 3 (OOD) | 42%                       | 26%                 |
>
> - We further evaluate our method on image restoration tasks. Specifically, we consider three training sets: ISTD [3], SRD [4], and Rain1400 [5]. The first two correspond to shadow removal, while Rain1400 focuses on image deraining. For each task, we first train a specialist model and then use it to generate pseudo-labeled images. The generalist is subsequently trained on the union of the original training data and the pseudo-labeled data. We report performance using peak signal-to-noise ratio (PSNR) and structural similarity index (SSIM), where higher values indicate better restoration quality.
>
>     | Task         | Semi-supervised learning  | Multi-task Baseline |
>     |---------------|--------------------------:|--------------------:|
>     | ISTD      | 27.88 / 0.950                      | 29.44 / 0.955                 |
>     | SRD       | 28.96 / 0.932                     | 29.14 / 0.939                 |
>     | Rain1400 |    30.31 / 0.913                  | 30.29 / 0.912                 |
>
> As shown in the table, our semi-supervised method is slightly worse than the multi-task baseline on ISTD and SRD, and performs comparably on Rain1400. Due to time constraints, we were only able to train the models for a limited number of iterations and explore a small set of hyperparameter configurations. We will continue refining these settings and update the results during the discussion stage.
>
> [1] Wegel, Tobias, et al. "On the sample complexity of semi-supervised multi-objective learning." arXiv preprint arXiv:2508.17152 (2025).
>
> [2] Tripuraneni, Nilesh, Michael Jordan, and Chi Jin. "On the theory of transfer learning: The importance of task diversity." Advances in neural information processing systems 33 (2020): 7852-7862.
>
> [3] Jifeng Wang, Xiang Li, and Jian Yang. Stacked Conditional Generative Adversarial Networks for Jointly Learning Shadow Detection and Shadow Removal. In Proc. CVPR, pages 1788–1797, 2018
>
> [4] Qu, L.; Tian, J.; He, S.; Tang, Y.; and Lau, R. W. 2017. Deshadownet: A multi-context embedding deep
> network for shadow removal. In CVPR, 4067–4075.
>
> [5] Fu X, Huang J, Zeng D, Huang Y, Ding X, and Paisley J. Removing rain from single images via a deep detail network. In IEEE CVPR 2017.

---

> > ### Author Rebuttal · Reviewer_V9TH · 2026-04-03
> >
> > I appreciate the theoretical rigor of the work, however I am not fully convinced of the limited experiments in the original submission. The additional experiments demonstrate improvement on the baseline only on a similar robot manipulation task as the one already included before.
> >
> > I encourage the authors to improve their experimental contributions when more time is available and strongly consider resubmission. I will keep my score as is.

---

> > > ### Author Response · Authors · 2026-04-06
> > >
> > > We thank the reviewer again for the helpful feedback, which motivated us to further strengthen the empirical evaluation. In response, we have added experiments on an image inpainting task.
> > >
> > > Specifically, we evaluate our method on CelebA-HQ [1] using three representative mask types from RePaint [2]: Wide, Face Mask, and Alternative Lines. The goal is to reconstruct the original image from the masked input. For each specialist model, we use 10k labeled images for training and generate pseudo-targets for the remaining 18k masked images using the trained specialist. The proposed semi-supervised generalist is then trained on the union of the labeled and pseudo-targeted data, resulting in 84k training images in total, while the multi-task baseline is trained on 30k labeled images. Both models are trained for 100k iterations and evaluated on 2,000 test images per mask type using Peak Signal-to-Noise Ratio (PSNR) and Structural Similarity Index Measure (SSIM), where higher values indicate better reconstruction quality.
> > >
> > >   The results are summarized below:
> > >
> > >   | Mask | Ours (Semi-Supervised Learning) | Multi-Task Baseline |
> > >   |---|---:|---:|
> > >   | Alt. Lines | **23.495 / 0.784** | 23.078 / 0.767 |
> > >   | Face Mask | **21.994 / 0.742** | 21.726 / 0.729 |
> > >   | Wide | **21.450 / 0.736** | 21.227 / 0.727 |
> > >
> > > As shown above, our method consistently outperforms the baseline across all metrics and mask types. These results demonstrate that our framework effectively leverages unlabeled data to improve reconstruction quality in inpainting, providing strong evidence of its generalizability beyond the tasks previously explored.
> > >
> > > We hope these additional results help clarify the empirical strength of our method, and we respectfully ask the reviewer to reconsider the decision in light of this expanded evidence.
> > >
> > > **References**
> > >
> > > [1] Karras, Tero, et al. "Progressive growing of gans for improved quality, stability, and variation." arXiv preprint arXiv:1710.10196 (2017).
> > >
> > >
> > > [2] Lugmayr, Andreas, et al. "Repaint: Inpainting using denoising diffusion probabilistic models." Proceedings of the IEEE/CVF conference on computer vision and pattern recognition. 2022.

---

### Decision · Program_Chairs · 2026-04-30

**Decision:**

Accept (regular)

**Comment:**

This paper considers multi-objective learning for diffusion models based on semi-supervised learning formulations.

### Strengths

- The multi-objective, semi-supervised setting is natural for real deployments of diffusion models, and the paper gives it a clear theoretical formulation.
- The main contribution is technical. Reviewers generally agreed that adapting multi-objective semi-supervised learning ideas to score-based diffusion models, and deriving guarantees in this setting, is nontrivial and original.
- A key positive aspect is the theoretical insight that the paired-sample complexity depends on the specialist model classes rather than the much larger generalist class.

### Weaknesses

- The realizability assumptions limits how directly theories apply to realistic model classes.
- The experimental section in the original submission was limited for several reviewers.
- Some reviewers raised questions on technical derivations, light-tailed assumptions, proof steps, and the role of unlabeled data and pseudo-samples in the bounds.

### Rebuttal and Remaining Concerns

The rebuttal addressed a fraction of the concerns. The authors clarified technical assumptions and proof details, explained the role of unlabeled data, and most importantly added substantially more empirical evidence, including both robotics and image tasks. These additions fully addressed the concerns from the two supportive reviewers.

At the same time, the rebuttal did not completely persuade the two weak-reject reviewers. The main remaining concerns are that the assumptions remain somewhat idealized and that the empirical support, while improved, is still not as broad or mature as one might hope for an ICML paper.

Overall, this is a borderline paper, with a slight accept lean because the theoretical contribution is novel. However, it should also be mentioned that there are non-negligible necessary revisions to fix the issues in the tail assumption, add complete derivations, and incorporate the additional experiments into the manuscript.